# The E3 ubiquitin ligase adaptor KLHL8 targets ZAR1 to regulate maternal mRNA degradation in oocytes

Huizhen Fan[1], Ruyi Liu [ID][1], Ran Yu[1], Biaobang Chen [ID][2], Qiaoli Li[1], Jian Mu[1], Weijie Wang [ID][3], Tianyu Wu [ID][1], Lin He[4], Lei Wang [ID][1,5]✉, Qing Sang [ID][1]✉ & Zhihua Zhang [ID][1]✉

## Abstract

**Maternal protein homeostasis and timely degradation of maternal mRNAs are essential for meiotic cell-cycle progression and subsequent embryonic development, but the mechanisms of maternal protein degradation are poorly understood. Here, we show that KLHL8, a substrate adaptor of Cullin-RING E3 ubiquitin ligases, is highly expressed in mouse oocytes and co-localizes with mitochondria. Oocyte-specific deletion of *Klhl8* causes oocyte maturation defects and female infertility. ZAR1, an RNA binding protein that is required for mitochondria-associated ribonucleoprotein domain (MARDO) dissolution, is specifically recognized and degraded by KLHL8-mediated ubiquitination. In *Klhl8*-deficient oocytes, ZAR1 accumulation causes abnormal MARDO and mitochondria clustering, correlating with impaired maternal mRNA decay. Supplementation with exogenous *Klhl8* mRNA rescues the degradation of ZAR1 and the dissolution of the MARDO in *Klhl8*$^{oo-/-}$ oocytes. Taken together, our study shows that KLHL8 mediates the ubiquitination and degradation of ZAR1, thus regulating maternal mRNA clearance during oocyte maturation. These findings provide new insights into the roles of the ubiquitin proteasome system during oocyte maturation and establish an interaction network between ubiquitination modification, RNA binding proteins, and maternal mRNA.**

**Keywords** KLHL8; Ubiquitination; Oocyte Maturation; ZAR1; Maternal mRNA Decay
**Subject Categories** Development; Post-translational Modifications & Proteolysis; RNA Biology

## Introduction

In mammals, fully grown oocytes are arrested at the germinal vesicle (GV) stage, at which time large amounts of mRNAs and proteins are accumulated and stored in the cytoplasm of oocytes (Wan et al, 2023). During the process of oocyte maturation, the majority of these maternal mRNAs undergo programmed decay, which is essential for fertilization and the subsequent oocyte to embryo transition. Several studies have shown that mRNA clearance is mediated by maternal proteins like the BTG4-CCR4–NOT complex (Soeda et al, 2023; Yu et al, 2016; Liu et al, 2016; Zhao et al, 2020). Apart from the clearance of maternal mRNAs, a group of maternal proteins are also decreased from the GV to metaphase II (MII) stage, including RNA splicing proteins, translational regulators, and several transporters (Wang et al, 2010; Sun et al, 2023; Zhang et al, 2023a), and this degradation is essential for oocyte maturation and subsequent embryonic development. For example, many RNA binding proteins are significantly decreased during meiosis, like ZAR1, ZAR2, YBX2, and PATL2, which are responsible for maternal mRNA stability and metabolism (Rong et al, 2019; Wu and Fan, 2022; Zhang et al, 2023c, 2023b). Thus, a better understanding of the mechanism of protein degradation in oocytes is of great importance.

The ubiquitin-proteasome system has been extensively investigated in terms of its ability to regulate protein degradation, which is mediated by ubiquitin-activating enzymes (E1s), ubiquitin-conjugating enzymes (E2s), and ubiquitin-ligase enzymes (E3s) (Kisielnicka et al, 2018; Karabinova et al, 2011). Among these, E3 ubiquitin ligases determine the precise substrate specificity of ubiquitination (DeRenzo and Seydoux, 2004; Suzumori et al, 2003). Multimeric E3 ubiquitin ligases are composed of a cullin-containing core and a variable substrate-recognition adaptor, and the specificity of E3 ligases is determined by adaptors such as F-box proteins and WD40 repeat-containing proteins, which bind one or a small number of targets in the ubiquitination reaction (DeRenzo and Seydoux, 2004). For example, the degradation of the stem-loop binding protein is regulated by the adaptor FBXO30 (which is one of the F-box protein) in mouse oocytes (Jin et al, 2019). DCAF1, a WD40 repeat-containing protein, targets protein phosphatase 2A degradation to control oocyte meiotic maturation (Yu et al, 2015), and DCAF13 deletion prevents oocyte meiosis, arrests follicle development, and results in female infertility due to the failure of PTEN degradation (Zhang et al, 2020). Despite the critical role of adaptors in regulating the specificity of protein degradation, our understanding of adaptor proteins and their corresponding targets in oocytes is quite limited.

[1]Institute of Pediatrics, Children's Hospital of Fudan University, the Institutes of Biomedical Sciences, The State Key Laboratory of Genetic Engineering, Fudan University, Shanghai, China. [2]NHC Key Lab of Reproduction Regulation (Shanghai Institute for Biomedical and Pharmaceutical Technologies), Fudan University, Shanghai, China. [3]International Peace Maternity and Child Health Hospital, School of Medicine, Shanghai Jiao Tong University, 200030 Shanghai, China. [4]Bio-X Center, Key Laboratory for the Genetics of Developmental and Neuropsychiatric Disorders, Ministry of Education, Shanghai Jiao Tong University, 200030 Shanghai, China. [5]Shanghai Academy of Natural Sciences (SANS), Fudan University, 200032 Shanghai, China. ✉E-mail: wangleiwanglei@fudan.edu.cn; sangqing@fudan.edu.cn; zhihuazhang_@fudan.edu.cn

The Kelch-like (KLHL) family, with a total of 42 members, is an important family of substrate adaptors that combine with Cullin3 to form various Cullin-RING E3 ubiquitin ligases (CRL3s) (Ye et al, 2022; Asmar et al, 2023) that are involved in a variety of cellular mechanisms (Dhanoa et al, 2013). Many KLHLs have been shown to ubiquitinate target substrates that regulate physiological and pathological processes in somatic cells (Zhou et al, 2024). Both KLHL2 and KLHL3 are reported to target WNK ubiquitination to regulate electrolyte homeostasis (Takahashi et al, 2013; Shibata et al, 2013). KLHL9, KLHL13, KLHL21, and KLHL22 were initially identified to control the dynamic behavior of Aurora B on mitotic chromosomes and thereby coordinate mitotic progression and the completion of cytokinesis (Sumara et al, 2007; Maerki et al, 2009). However, whether KLHL family proteins are involved in protein degradation in oocytes, and if so, what are the specific substrates remain to be elucidated.

In this study, we focused on a KLHL member protein, KLHL8, that is highly expressed in mouse oocytes, and oocyte-specific deletion of *Klhl8* caused MI arrest and female infertility, suggesting the essential role of KLHL8 in oocyte meiotic maturation. Mechanistic studies indicated that KLHL8 deficiency caused the accumulation of RNA-binding proteins such as ZAR1, which then impaired MARDO dissolution and maternal mRNA decay. Our findings suggest a vital role for KLHL8 in facilitating the degradation of specific proteins, thus regulating meiotic progression and ensuring oocyte quality.

## Results

### KLHL8 is highly expressed in mouse oocytes

To screen the major KLHL members that play critical roles in meiosis process, we first assessed the expression of the KLHL gene family during oocyte development. The transcription level of *Klhl8* is was the highest among the KLHL family members during oocyte maturation in our in-house database (Fig. EV1A) (Zhao et al, 2022).

To investigate the functions of KLHL8 in oocyte maturation and female fertility, we determined the expression pattern of *Klhl8* in mouse oocytes, early embryos, and diverse somatic tissues. The real-time quantitative reverse transcription PCR (qRT-PCR)

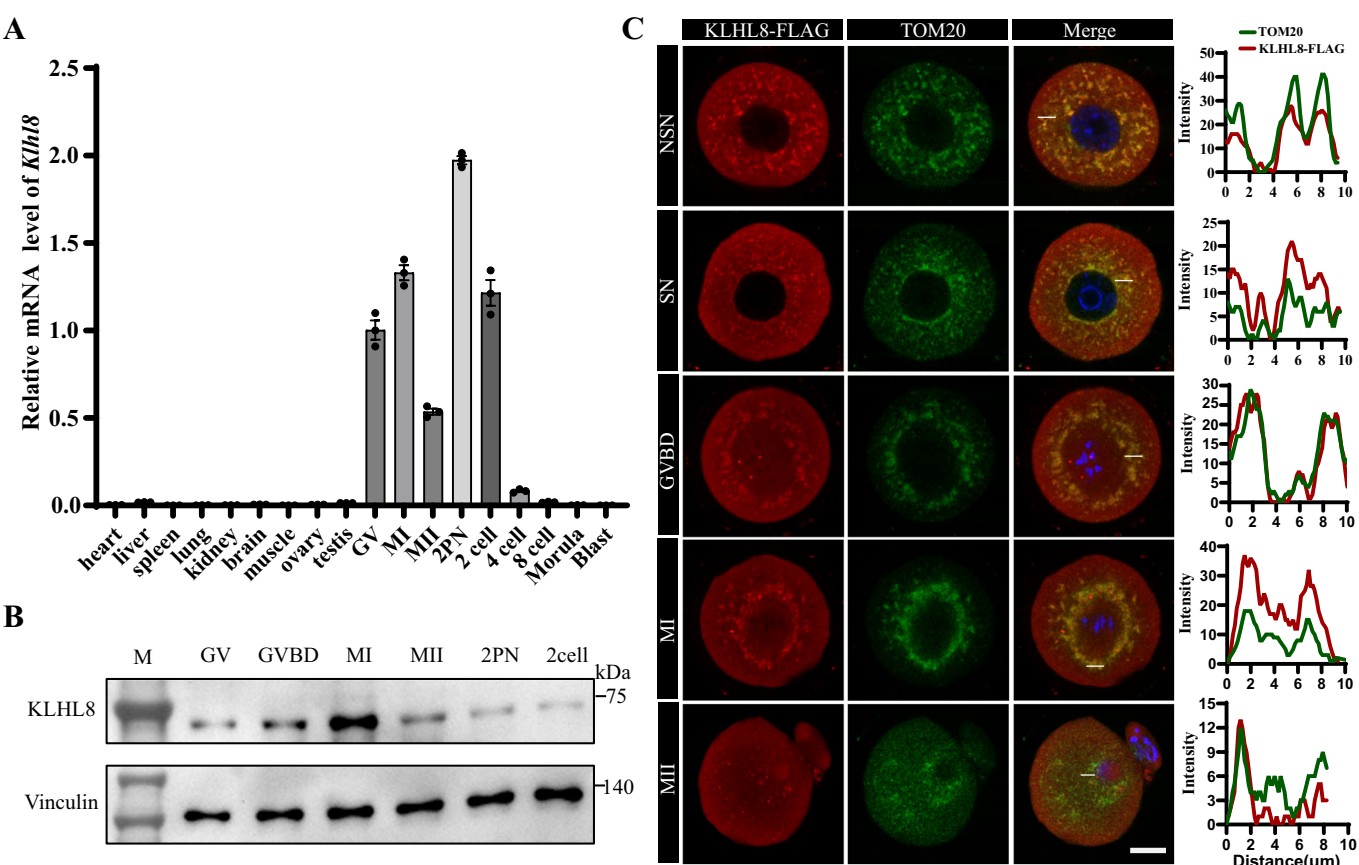

**Figure 1. Expression pattern and localization of KLHL8 in oocytes.**

(A) qRT-PCR results for *Klhl8* expression in mouse oocytes, pre-implantation embryos, and somatic tissues. β-actin was used as the internal control. Data are presented as mean ± SEM (n = 3). (B) Immunoblotting results showing KLHL8 protein levels in oocytes during meiotic maturation and embryos at the indicated developmental stages. Vinculin was used as the loading control. (C) Representative immunofluorescence images of mouse oocytes at the indicated growth stages. Red, KLHL8-FLAG; Green, TOM20 (used to label the mitochondria). Scale bar, 20 μm. The intensity profiles of the fluorescence of KLHL8 and TOM20 are shown along the white lines. Source data are available online for this figure.

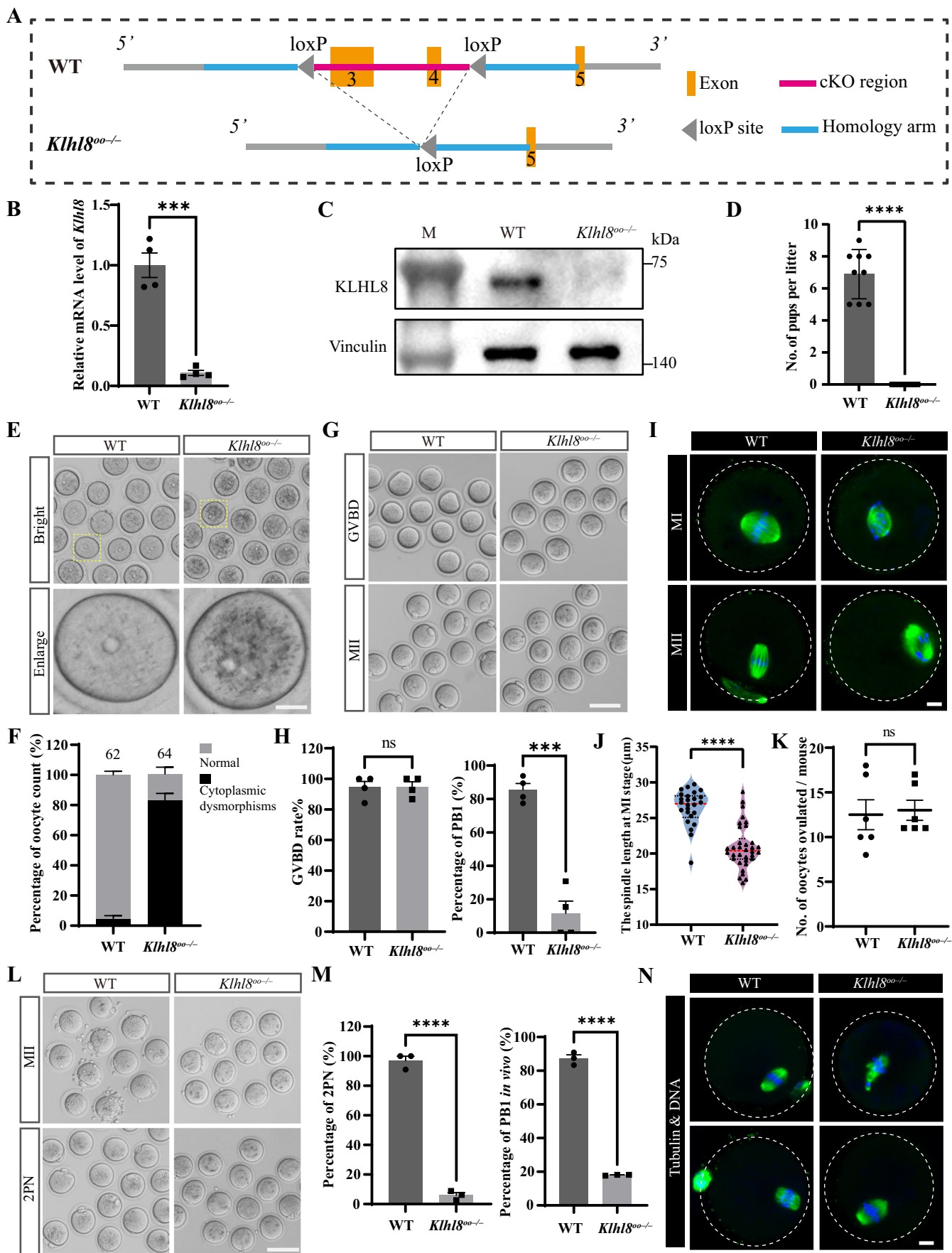

**Figure 2. Deletion of *Klhl8* in mouse oocytes causes female infertility characterized by oocyte maturation defects.**

(A) Strategy for creating oocyte-specific *Klhl8* conditional knockout (cKO) mice. (B) qRT-PCR results of *Klhl8* expression in GV oocytes of WT and *Klhl8*$^{oo-/-}$ females. β-actin was used as the internal control. Data are presented as mean ± SEM ($n = 3$). Statistical significance was determined using an unpaired two-tailed $t$ test: \*\*\*$P = 0.0001$. (C) Immunoblotting confirmation that KLHL8 was deleted in *Klhl8*$^{oo-/-}$ oocytes. Vinculin was used as the loading control. (D) Numbers of pups per litter during the indicated time windows of the fertility test. Data are presented as mean ± SEM ($n = 9$). Statistical significance was determined using an unpaired two-tailed $t$ test: \*\*\*\*$P = 4 \times 10^{-10}$. (E) Representative images of GV oocyte morphology in WT and *Klhl8*$^{oo-/-}$ mice at 7–8 weeks. Scale bar, 100 μm. (F) The rates of cytoplasmic dysmorphisms in GV oocytes collected from WT (62 oocytes) and *Klhl8*$^{oo-/-}$ (64 oocytes) female mice. Data are presented as mean ± SEM ($n = 3$). (G) Representative images of GVBD-stage oocytes and in vitro-matured oocytes from WT and *Klhl8*$^{oo-/-}$ female mice. Scale bar, 100 μm. (H) The GVBD rates and the percentages of oocytes with a first polar body (PB1) from WT and *Klhl8*$^{oo-/-}$ females in vitro. Data are presented as mean ± SEM ($n = 4$). Statistical significance was determined using an unpaired two-tailed $t$ test: n not significant, \*\*\*$P = 0.0001$. (I) Identification of the spindles in WT and *Klhl8*$^{oo-/-}$ oocytes after culturing for 8 h or 14 h in M2 medium. Scale bar, 10 μm. β-Tubulin and Hoechst staining were used to label the spindle and DNA, respectively. The dashed line demarcates the oocytes. (J) Quantification of the spindle length at the MI stage (WT, $n = 25$; *Klhl8*$^{oo-/-}$, $n = 33$). Data are presented as mean ± SEM. Statistical significance was determined using an unpaired two-tailed $t$ test: \*\*\*\*$P = 4 \times 10^{-10}$. (K) The number of oocytes ovulated from WT and *Klhl8*$^{oo-/-}$ female mice. Data are presented as mean ± SEM ($n = 6$). Statistical significance was determined using an unpaired two-tailed $t$ test: ns = not significant. (L) Representative images of oocytes ovulated in vivo and 2PN zygotes after fertilization in WT and *Klhl8*$^{oo-/-}$ female mice. Scale bar, 100 μm. (M) The percentages of PB1 at 14 h after ovulation induction and the percentages of 2PN zygotes after fertilization. Data are presented as mean ± SEM ($n = 3$). Statistical significance was determined using an unpaired two-tailed $t$ test: \*\*\*\*$P = 1 \times 10^{-5}$, \*\*\*\*$P = 6 \times 10^{-6}$. (N) Immunofluorescence of ovulated oocytes from WT and *Klhl8*$^{oo-/-}$ female mice. Scale bar, 10 μm. β-Tubulin and Hoechst staining were used to label the spindle and DNA. The dashed line demarcates the oocytes. Source data are available online for this figure.

showed that *Klhl8* mRNA was highly expressed specifically in oocytes (Fig. 1A). Western blot assay also showed a trend in which the KLHL8 protein level was highest in metaphase I (MI) oocytes, with the abundance decreasing during meiotic maturation and becoming only mildly detectable in 2 pronuclear (2PN) embryos (Figs. 1B and EV1B). We then overexpressed KLHL8-FLAG in oocytes and found that KLHL8 was co-localized with the mitochondria in oocytes (Fig. 1C). KLHL8 was highly concentrated at mitochondria in the GV oocytes containing non-surrounded nucleolus (NSN) and MI oocytes, but mitochondrial localization was almost completely absent in MII oocytes. This co-localization was not observed in somatic cells (Appendix Fig. S1), which indicated a specific role of KLHL8 in oocytes.

## Deletion of *Klhl8* in mouse oocytes results in female infertility characterized by MI arrest

To investigate the functions of *Klhl8* in female fertility in vivo, we generated an oocyte-specific *Klhl8* conditional knockout mouse (*Klhl8*$^{oo-/-}$) using a CRISPR/Cas9 genome editing strategy. A stretch of 2593 nucleotides, including exons 3 and 4, of the *Klhl8* gene was deleted (Fig. 2A), and qRT-PCR and immunoblotting confirmed that the *Klhl8* mRNA and KLHL8 protein were knocked out in *Klhl8*$^{oo-/-}$ oocytes (Fig. 2B,C). The *Klhl8*$^{oo-/-}$ mice were viable and healthy, and the size and histology of the ovary showed no obvious difference compared to control mice (Appendix Fig. S2A,B). Nevertheless, fertility testing indicated that the *Klhl8*$^{oo-/-}$ female mice were completely infertile (Fig. 2D). We then collected GV oocytes from the ovaries, and most of them showed obvious dysmorphisms and were filled with cytoplasmic granules (Fig. 2E,F). Although germinal vesicle breakdown (GVBD) rate of oocytes was not affected (Fig. 2G,H), most *Klhl8*$^{oo-/-}$ oocytes (85.1%) could not extrude the first polar body when they matured in vitro (Fig. 2G–J). Immunoblotting demonstrated that the activity of MPF was not affected after *Klhl8* deletion (Appendix Fig. S3A,B). In vivo super-ovulation indicated that the number of oocytes was comparable between the two groups (Fig. 2K), but the majority of *Klhl8*$^{oo-/-}$ ovulated oocytes showed oocyte maturation arrest (Fig. 2L,M). Immunofluorescence staining showed that the oocytes were arrested at the MI stage and had abnormal spindle assembly or disordered chromosomal arrangement (Fig. 2I,J,N). Although a small proportion of *Klhl8*$^{oo-/-}$ oocytes could extrude the first polar body, these still failed to be fertilized (Fig. 2L,M). Most of them showed absent spindles, which entered the first polar body without division and exhibit abnormal morphology. A small portion of oocytes had obvious cytoplasmic fragmentation and abnormal spindle morphology that are not properly separated (Fig. EV2A,B). Together, these results suggest that *Klhl8* plays an essential role in oocyte maturation and female fertility.

## KLHL8 deficiency impedes the degradation of RNA binding proteins

Because KLHL8 is a ubiquitination E3 ligase adaptor, we tested whether KLHL8 is involved in protein degradation during oocyte maturation. Proteomics analysis was performed in WT and *Klhl8*$^{oo-/-}$ GV oocytes (Fig. 3A; Dataset EV1). Compared with oocytes derived from WT mice, 445 and 234 proteins were increased or decreased by >1.4-fold in the GV oocytes of *Klhl8*$^{oo-/-}$ mice (Fig. 3A). Gene ontology (GO) enrichment showed that the increased proteins in *Klhl8*$^{oo-/-}$ oocytes were mostly clustered in terms related to mRNA dynamics such as mRNA processing, RNA splicing, etc., (Fig. 3B), and most of them were RNA binding proteins (Fig. 3C). These results indicated that KLHL8 participates in the degradation of proteins related to mRNA homeostasis. Among the elevated proteins, we focused on the RNA binding protein ZAR1, which co-localizes with mitochondria and plays important roles in oocyte maturation. Western blotting confirmed that *Klhl8* deletion sharply increased the protein level of ZAR1 (Fig. 3D,E). qRT-PCR results showed that *Klhl8* deletion did not affect the level of *Zar1* transcripts (Fig. 3F). There is no difference in translation efficiency between WT and *Klhl8*$^{oo-/-}$ oocytes (Fig. EV3), indicating that the accumulation of ZAR1 was due to the protein degradation. It has been reported that ZAR1 is essential for the assembly of the MARDO, a mitochondria-associated membraneless compartment that stores maternal mRNAs in oocytes (Cheng et al, 2022). YBX2 and DDX6 are also involved in the MARDO, but the protein levels of DDX6 and YBX2 showed no obvious differences in *Klhl8*$^{oo-/-}$ oocytes (Fig. 3G,H), suggesting that the targets of KLHL8 are selective.

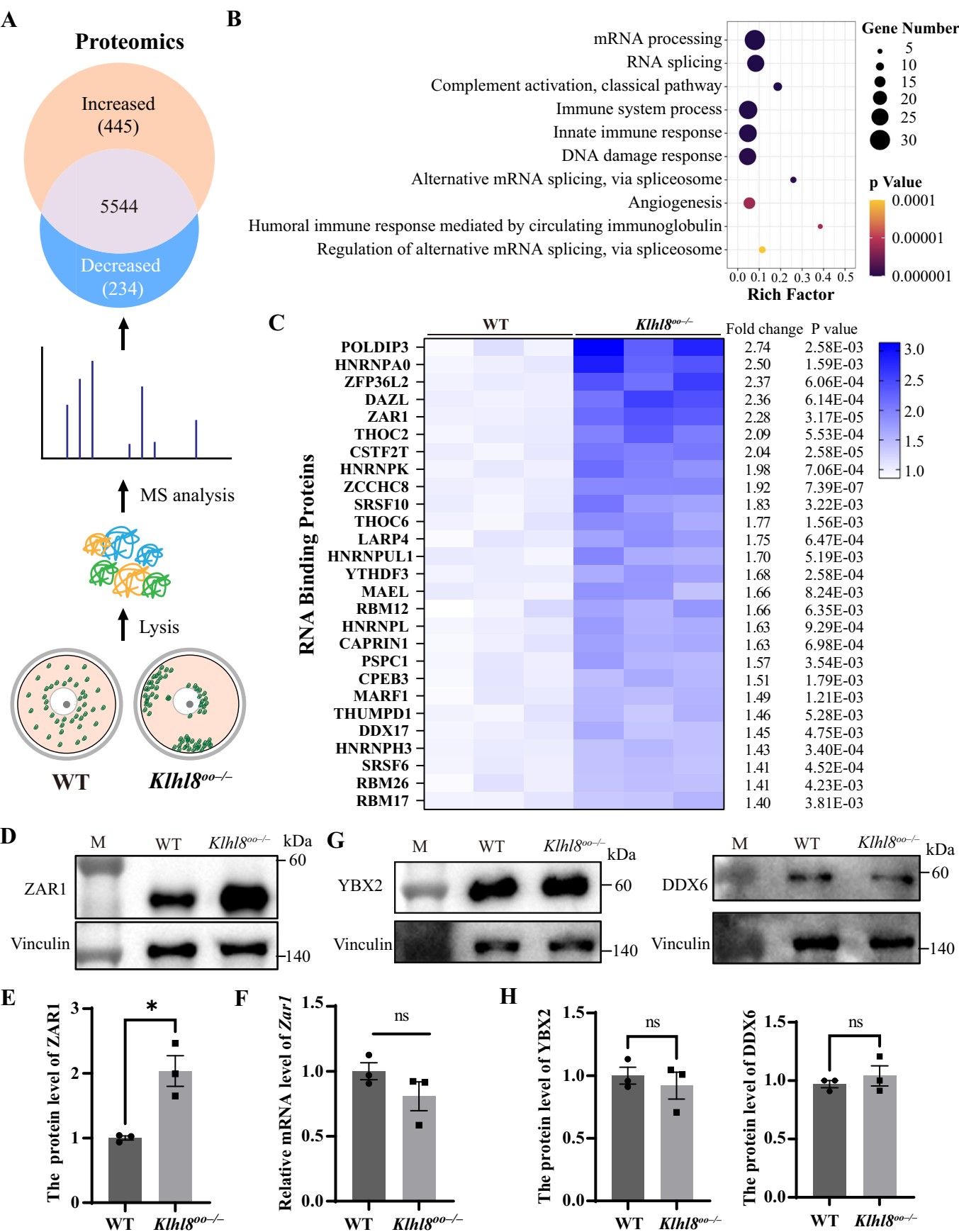

◀ **Figure 3.  *Klhl8* deletion causes increased protein levels of RNA-binding proteins.**

(A) Venn diagrams showing the number of increased or decreased proteins in the proteomics analysis. Proteins that increased or decreased by >1.4-fold in *Klhl8^oo−/−^* oocytes are shown in orange or blue, respectively. (B) Bubble charts showing GO enrichment analysis of the genes coding increased proteins in *Klhl8^oo−/−^* GV oocytes. *P* values were calculated using Fisher's exact test as implemented in DAVID. (C) Heatmap illustration showing the protein level of representative RNA binding proteins that are increased in *Klhl8^oo−/−^* GV oocytes. *P* values were calculated using *t* test. (D) Western blot results showing the ZAR1 protein level in WT and *Klhl8^oo−/−^* GV oocytes. Vinculin was used as the protein loading control. (E) Quantitative analysis of the ZAR1 protein level. Data are presented as mean ± SEM (*n* = 3). Statistical significance was determined using an unpaired two-tailed *t* test: *$P$ = 0.0121. (F) qRT-PCR results showing the ZAR1 transcription level in WT and *Klhl8^oo−/−^* GV oocytes. Data are presented as mean ± SEM (*n* = 3). Statistical significance was determined using an unpaired two-tailed *t* test: ns not significant. (G) Western blot results showing the expression of other MARDO-related proteins in WT and *Klhl8^oo−/−^* GV oocytes. Vinculin was used as the protein loading control. (H) Quantitative analysis of the YBX2 and DDX6 protein level. Data are presented as mean ± SEM (*n* = 3). Statistical significance was determined using an unpaired two-tailed *t* test: ns not significant. Source data are available online for this figure.

## KLHL8 binds to ZAR1 for ubiquitination and proteasome degradation

Based on the above results, we hypothesized that KLHL8 targets ZAR1 degradation via the proteasome pathway. Co-immunoprecipitation in HeLa cells showed that KLHL8 interacted with ZAR1 clearly but had a very weak interaction with YBX2 (Fig. 4A). To determine the binding region of KLHL8 and ZAR1, we created truncated versions of ZAR1 and found that deletion of either of two segments (Δ218-289 and Δ290-361) prevented the binding with KLHL8 (Fig. 4B). This result indicated that the ZAR1 C-terminal domain (the zinc-binding domain) is necessary for KLHL8 binding. KLHL8 possesses a BTB/POZ domain, a BACK domain, and six Kelch motifs (Zhou et al, 2024), and we established the BTB and BACK domain truncations of KLHL8. Of the two, the BACK domain was found to be required for the KLHL8-ZAR1 interaction (Fig. 4C). These results further confirmed that the interaction between KLHL8 and ZAR1 is specific.

Next, we explored whether ZAR1 degradation was controlled by KLHL8 via the ubiquitin-proteasome system. We added MG132 (a proteasome inhibitor) for different duration after overexpressing ZAR1 in HeLa cells, and ZAR1 gradually accumulated over time (Fig. EV4A,B), suggesting that the stability of ZAR1 is regulated by a proteasome-mediated pathway. Moreover, co-expressed with KLHL8 could promote the degradation of ZAR1 in HeLa cells, while adding MG132 could prevent the decreasing (Fig. 4D,E). In vivo ubiquitination assays also showed that co-transfection of KLHL8 could promote the ubiquitination of ZAR1 (Fig. 4F). KLHL8 mutants with BACK domain deletion lost the ability to promote the degradation and ubiquitination of ZAR1 (Fig. EV4C–E). Collectively, these results demonstrated that KLHL8 binds to ZAR1 and promotes its ubiquitination and proteasome degradation.

## KLHL8 deficiency induces mitochondrial and MARDO aggregation

Cheng et al (2022) reported that ZAR1 can promote MARDO coalescence and mitochondrial clustering. The GV oocytes we collected exhibited obvious cytoplasmic dysmorphisms (Fig. 2E,F), and the dark and granular cytoplasm was usually caused by abnormal organelle distribution (Balaban et al, 2008; Udagawa and Ishihara, 2020; Udagawa et al, 2014). We therefore hypothesized that KLHL8 modulates the MARDO's behavior by regulating the degradation of ZAR1. Immunofluorescence staining showed that the mitochondria in *Klhl8^oo−/−^* oocytes formed large, irregularly distributed aggregates (Fig. 5A), and all oocytes with *Klhl8* deletion shifted from a uniform distribution to a pattern in which the mitochondria were accumulated around the nucleus and in the subcortical region. Transmission electron microscopy also showed an obvious mitochondrial clustering in *Klhl8^oo−/−^* oocytes (Fig. 5B). Immunofluorescence staining of ZAR1 and YBX2 showed that the MARDO formed severely abnormal clusters throughout the cytoplasm (Fig. 5C; Appendix Fig. S4), and the mitochondrial cluster index confirmed the high degree of clustering of the MARDO (Fig. 5D). However, oocyte deletion of *Klhl8* had no effect on mitochondrial membrane potential (MMP), reactive oxygen species (ROS) or ATP levels (Appendix Fig. S5A–E). Collectively, these results indicate that KLHL8 modulates the MARDO's behavior by regulating the degradation of ZAR1.

## Supplementation with *Klhl8* mRNA decreases the protein level of ZAR1 and rescues mitochondrial distribution defects

To further confirm the crucial role of KLHL8 in degrading ZAR1 and regulating MARDO homeostasis in oocytes, we tried to rescue the abnormal phenotype in *Klhl8^oo−/−^* oocytes at the GV stage by supplying *Klhl8* mRNA. As predicted, the protein level of ZAR1 decreased in *Klhl8^oo−/−^* GV oocytes after exogenous *Klhl8* mRNA supplementation compared to noninjected and *Klhl8*-ΔBACK mRNA injection oocytes (Figs. 6A,B and EV4F). Consistent with the abnormality we observed in GV oocytes, the distribution of mitochondria was abnormally clustered at different stages in *Klhl8^oo−/−^* oocytes. The mitochondria were clustered around the MI spindle in the WT group but completely away from the spindle in *Klhl8^oo−/−^* oocytes. However, after exogenous *Klhl8* mRNA supplementation, mitochondria regained their normal distribution pattern in the GV, MI, and MII stages in *Klhl8^oo−/−^* oocytes (Fig. 6C). In the meantime, the MARDO (stained with ZAR1) was gradually dissolved after *Klhl8* mRNA injection at the GV stage (Fig. 6D,E). The PB1 rate was also significantly improved after *Klhl8* mRNA supplementation in *Klhl8^oo−/−^* oocytes (Fig. 6F,G). These results further validated the key role of KLHL8 in the ubiquitination-dependent degradation of ZAR1 and the dissolution of the MARDO.

## KLHL8 deficiency impairs maternal mRNA decay

It has been reported that the MARDO structure needs to be dissolved during oocyte maturation and that this is essential for the timely degradation of maternal mRNAs (Cheng et al, 2022). However, in *Klhl8^oo−/−^* oocytes, the MARDO was not dissolved but instead accumulated in aggregates. To verify whether this change

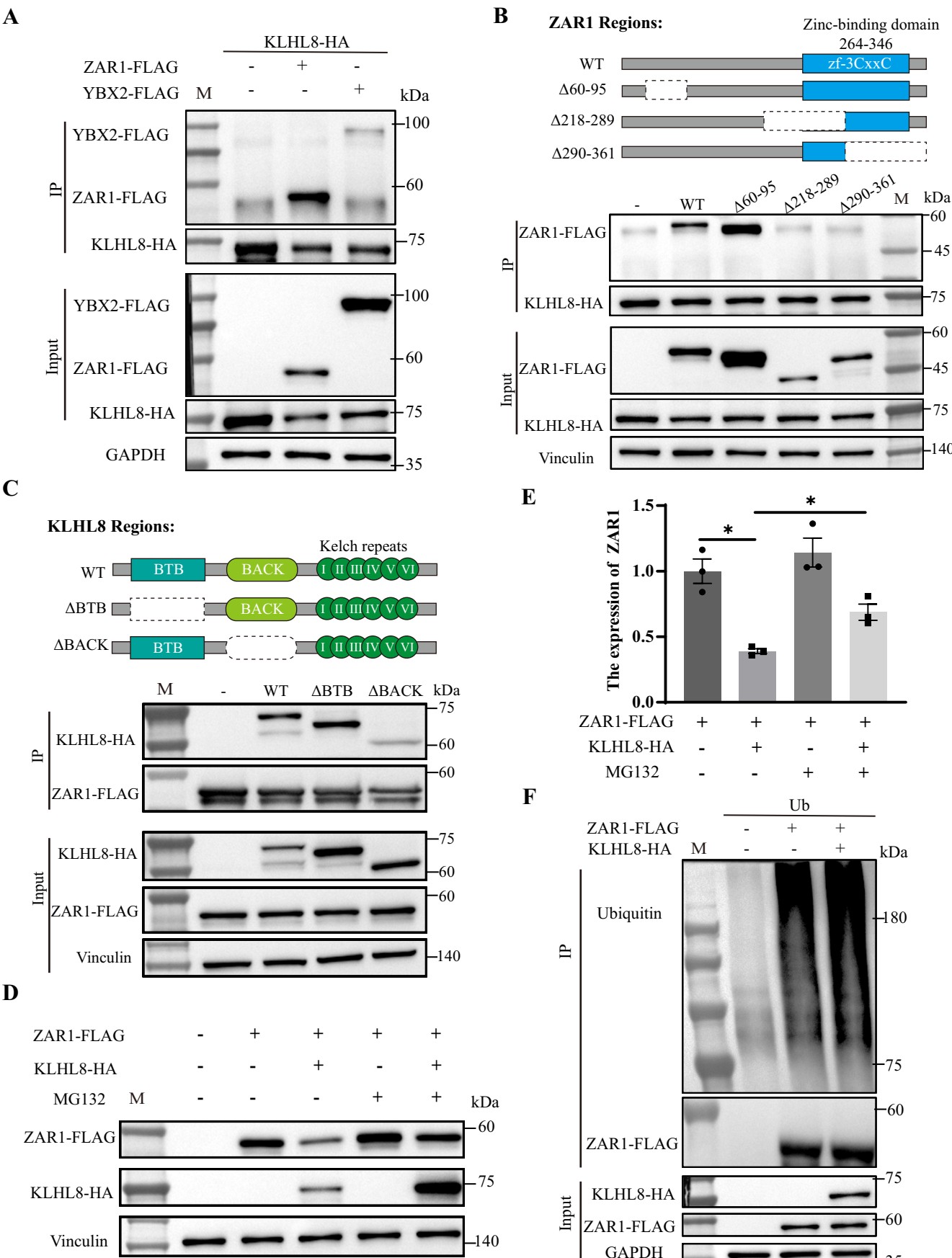

**Figure 4. KLHL8 targets ZAR1 degradation by promoting its ubiquitination.**

(A) Co-immunoprecipitation of KLHL8 with ZAR1 or YBX2 in cultured HeLa cells. (B) Co-immunoprecipitation showed the effect of ZAR1 domain on KLHL8-ZAR1 interaction. (C) Co-immunoprecipitation showed the effect of KLHL8 domain on KLHL8-ZAR1 binding. (D) The effect of KLHL8 on the ZAR1 protein level in cultured HeLa cells with or without MG132. (E) Quantitative analysis of ZAR1 protein level. Data are presented as mean ± SEM ($n = 3$). Statistical significance was determined using an unpaired two-tailed $t$ test: $*P = 0.02$, $*P = 0.042$. (F) FLAG IP of HeLa cells transfected with ZAR1-FLAG and with or without KLHL8-HA, followed by immunoblot for ubiquitin. Source data are available online for this figure.

impacted mRNA homeostasis, GV and MII oocytes from control and $Klhl8^{oo-/-}$ female mice were collected for RNA-sequencing (Dataset EV2). A total of 486 and 282 transcripts were upregulated and downregulated (fold change >3) in the $Klhl8^{oo-/-}$ GV stage oocytes, respectively (Fig. 7A, left panel). This effect became more apparent in ovulated MII oocytes, in which more genes were upregulated (1776) than downregulated (467; Fig. 7A, right panel). This demonstrated that a larger number of transcripts were accumulated in $Klhl8^{oo-/-}$ MII oocytes indicating a disruption of maternal mRNA clearance. In addition, 486 transcripts exhibited higher copy numbers in $Klhl8^{oo-/-}$ oocytes compared to WT oocytes at the GV stage (Fig. 7A, right panel), with half of these transcripts also exhibiting increased expression levels at the MII stage (Fig. 7B). The qRT-PCR assay further confirmed that the selected maternal transcripts were degraded in WT oocytes but accumulated in $Klhl8^{oo-/-}$ oocytes (Fig. 7C). Oligo(dT) staining also revealed that poly(A) mRNA clearance occurred during the transition from MI to MII. When $Klhl8$ was deleted, a significant fraction of MARDO-localized mRNAs was retained in MII oocytes (Fig. 7D,E). Collectively, these results suggest that KLHL8 is essential for ZAR1 degradation, which is crucial for promoting maternal mRNA decay during oocyte meiotic maturation.

## Discussion

In this study, we investigated the physiological functions and biochemical mechanisms of KLHL8 in regulating mouse oocyte meiotic maturation. Oocyte-specific deletion of $Klhl8$ caused oocytes MI arrest characterized by abnormal spindle assembly. We also showed that KLHL8 deficiency caused ZAR1 accumulation and induced mitochondrial and MARDO aggregation thus impairing maternal mRNA decay. This result highlighted KLHL8-promoted ubiquitination of ZAR1 in regulating maternal mRNA degradation and oocyte development (Fig. 8).

In addition to ZAR1, our proteomics analysis showed that KLHL8 deficiency also caused the accumulation of other important RNA-binding proteins that are involved in the regulation of oocyte development (Fig. 3C). These differentially expressed proteins might be responsible for the developmental retardation and infertility. DAZL and MARF1 were also accumulated in $Klhl8$ deleted oocytes (Figs. 3C and EV5A,B). DAZL, a key RNA-binding protein that has been reported to control maternal mRNA translation (Chen et al, 2011; Fukuda et al, 2018), is involved in the development of female germ cells, and overexpression of DAZL leads to pre-implantation developmental defects (Fukuda et al, 2018). MARF1 is essential for meiotic progression and female fertility. Mutations in $MARF1$ cause female infertility characterized by up-regulation of transcripts, defective cytoplasmic maturation, and meiotic arrest (Su et al, 2012). While, the protein level of

ZFP36L2 was obvious decreased in $Klhl8^{oo-/-}$ oocytes (Fig. EV5C,D). ZFP36L2 is important for transcriptional silence in oocytes (Chousal et al, 2018), and $Zfp36l2$ depletion results in oocyte maturation defects and aberrant spindle assembly (Sha et al, 2018). Sun et al performed a systematic quantitative proteomic analysis of GV, GVBD, and MII oocytes, and they showed that the proteomes could be partitioned into different clusters with distinct temporal patterns (Sun et al, 2023). We found obvious overlap between the proteins that accumulated in $Klhl8^{oo-/-}$ GV oocytes and those that are degraded when transitioning from the GV to MII stage (Data ref: Li et al, 2020; Li et al, 2020; Appendix Fig. S6A). Notably, GO enrichment analysis indicated that this set of proteins was enriched for regulation of RNA, such as mRNA processing and RNA splicing (Appendix Fig. S6B). This result further confirmed the necessity of KLHL8 for maternal mRNA metabolism. Therefore, more studies are required to confirm whether and how KLHL8 regulates the degradation of these maternal proteins to affect maternal mRNA metabolism and oocyte development potential.

KLHL8 co-localized with oocyte mitochondria, but it did not co-localize with mitochondria or ZAR1 in cultured HeLa cells (Appendix Figs. S1 and S7), and this indicates a specific role of KLHL8 in oocyte mitochondria. YBX2 is another member of the MARDO and is involved in regulating maternal mRNA stability (Zhang et al, 2023c), and it is worth noting that KLHL8 targeted the degradation of ZAR1 but had no effect on YBX2. $Zar1$ and $Zar2$ was reported partial functional redundancy in multiple studies (Wu and Fan, 2022; Rong et al, 2025). We performed western blot assay and found no difference in the protein abundance of ZAR2 after $Klhl8$ deletion (Fig. EV5E,F). It indicates that ZAR2 is unlikely to be a target of KLHL8. This suggests that KLHL8 has a selective effect on RNA binding proteins. Endolysosomal vesicular assemblies (ELVAs) are non-membrane-bound compartments composed of endolysosomes, autophagosomes, and proteasomes, which sequester and degrade aggregated proteins in oocytes (Zaffagnini et al, 2024). KLHL8 localized in the mitochondrial region and promoted proteins degradation through ubiquitination pathway, which is distinct from ELVA. The degradation activity of ELVAs is activated after oocyte maturation. However, KLHL8 is weakly expressed and localized in MII oocytes. We also detected the abundance of LSM14B and G3BP2 which had been confirmed not to be degraded by ELVAs. There was no significant difference in their protein levels between WT and $Klhl8^{oo-/-}$ oocytes (Fig. EV5G–J). Together, these findings suggest that the KLHL8-ZAR1-MARDO pathway is specific to oocytes.

The degradation of maternal mRNA during oocyte maturation mainly depends on the BTG4-CCR4–NOT complex (Soeda et al, 2023; Yu et al, 2016; Liu et al, 2016; Zhao et al, 2020). Encoding the CCR4–NOT catalytic subunit, $Cnot6l$ deletion disturbs the deadenylation and degradation of a subset of maternal mRNAs during mouse oocyte maturation causing MI arrest (Sha et al, 2018). BTG4, a

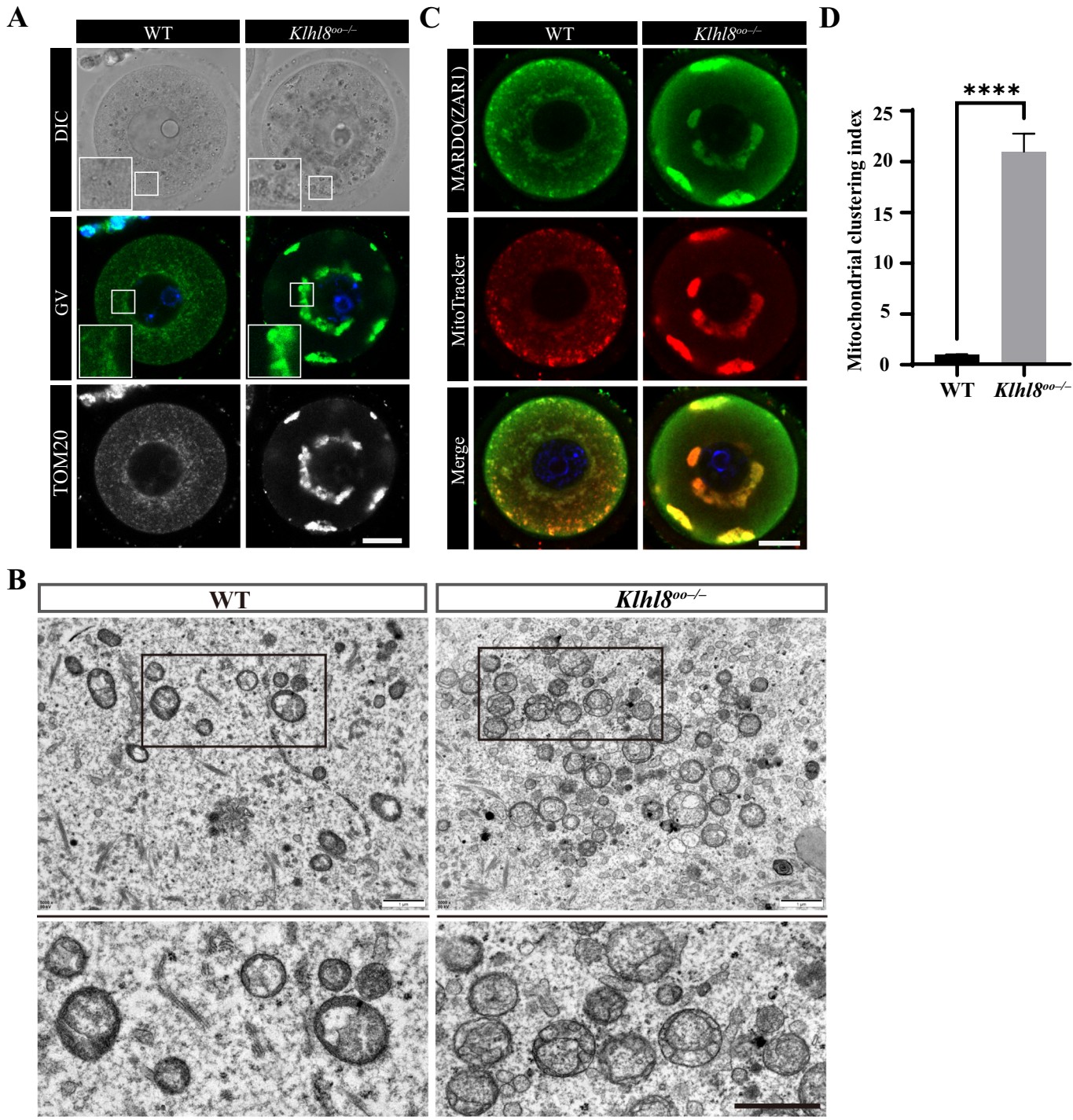

**Figure 5.** *Klhl8* deletion causes mitochondrial aggregates in oocytes.

(A) Subcellular distribution of the mitochondria in WT and *Klhl8^oo−/−^* oocytes. TOM20 was used to label the mitochondria. Insets are magnifications of the outlined regions. Scale bar, 20 μm. (B) Representative transmission electron microscopy images showing the distribution of mitochondria in WT and *Klhl8^oo−/−^* GV oocytes. The outlined regions are magnified at the bottom. Scale bar, 1 μm. (C) Representative immunofluorescence images of GV-stage oocytes collected from WT and *Klhl8^oo−/−^* female mice. Green, MARDO (ZAR1); red, mitochondria (MitoTracker). Scale bar, 20 μm. (D) Quantification of the mitochondrial clustering index in WT and *Klhl8^oo−/−^* GV oocytes (WT, $n = 37$; *Klhl8^oo−/−^*, $n = 37$). Data are presented as mean ± SEM. Statistical significance was determined using an unpaired two-tailed *t* test: ****$P = 1 \times 10^{-15}$. Source data are available online for this figure.

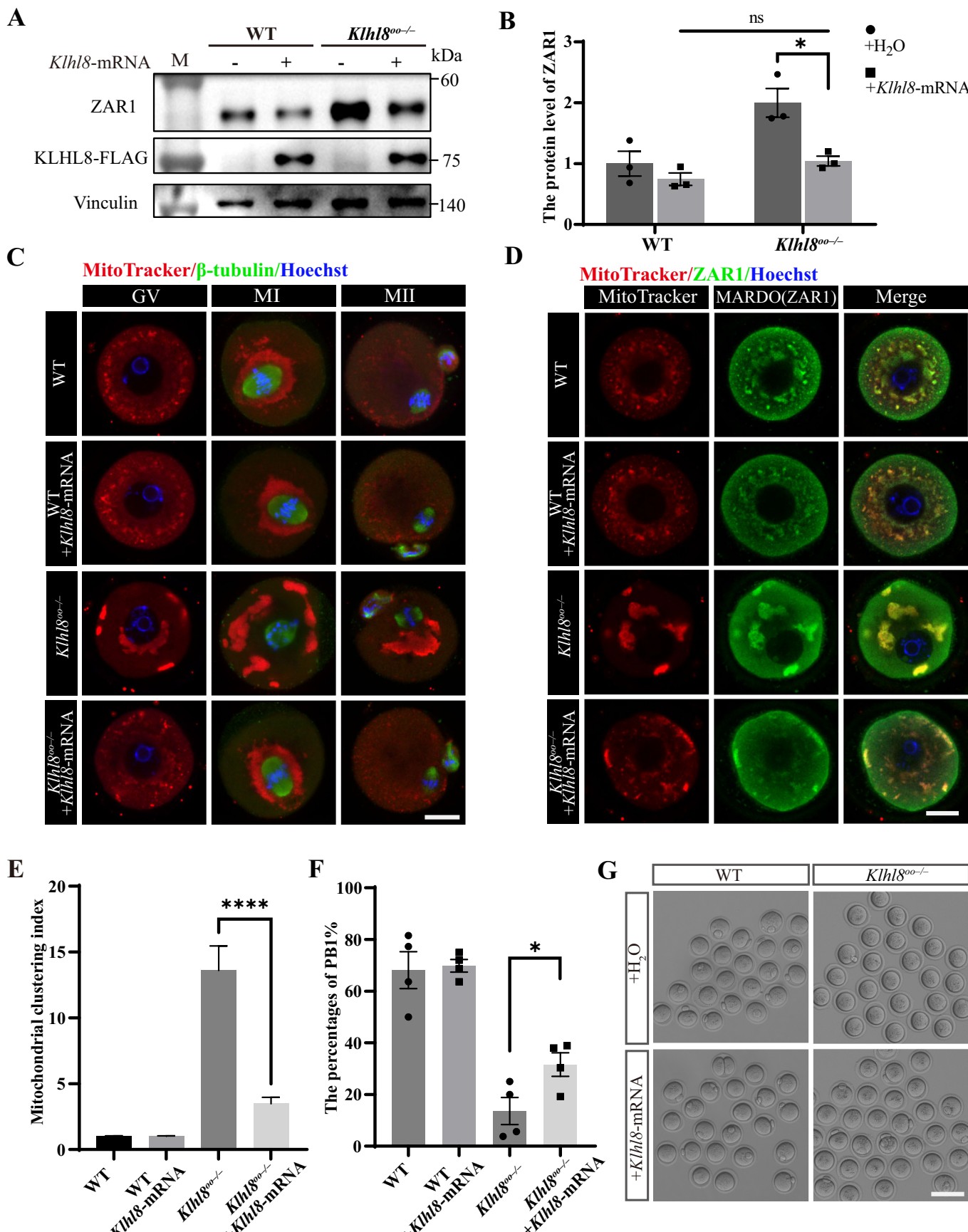

◄ **Figure 6.** ***Klhl8* mRNA injection decreases ZAR1 protein level and rescues mitochondrial distribution defects.**

(A) Immunoblotting for ZAR1 in WT and *Klhl8°°−/−* GV oocytes with or without *Klhl8* mRNA injection. Vinculin was used as the protein loading control. (B) Quantitative analysis of the ZAR1 protein level in WT and *Klhl8°°−/−* GV oocytes after *Klhl8* mRNA injection. Data are presented as mean ± SEM ($n = 3$). Statistical significance was determined using an unpaired two-tailed *t* test: ns not significant, *$P = 0.0185$. (C) Representative immunofluorescence images of the mitochondrial distribution in *Klhl8°°−/−* oocytes with or without *Klhl8* mRNA injection at different growth stages. Scale bar, 10 μm. (D) Representative immunofluorescence images of the MARDO with or without *Klhl8* mRNA injection in WT and *Klhl8°°−/−* GV oocytes. Green, MARDO (ZAR1); red, mitochondria (MitoTracker). Scale bar, 10 μm. (E) Quantification of the mitochondrial clustering index after *Klhl8* mRNA injection in WT and *Klhl8°°−/−* GV oocytes (WT, $n = 17$; WT + *Klhl8* mRNA, $n = 16$; *Klhl8°°−/−*, $n = 13$; *Klhl8°°−/−* + *Klhl8* mRNA, $n = 22$). Data are presented as mean ± SEM. Statistical significance was determined using an unpaired two-tailed *t* test: ****$P = 2 \times 10^{-7}$. (F) The percentages of PB1 in WT and *Klhl8°°−/−* mice with or without *Klhl8* mRNA injection. Data are presented as mean ± SEM ($n = 4$). Statistical significance was determined using an unpaired two-tailed *t* test: *$P = 0.0413$. (G) Representative images of GV oocytes and maturation oocytes from WT and *Klhl8°°−/−* mice with or without *Klhl8* mRNA injection. Scale bar, 100 μm. Source data are available online for this figure.

CCR4–NOT adaptor, has been reported to impair maternal mRNA decay (Yu et al, 2016). However, the degradation of these RNA-binding proteins in oocytes has not been well studied. Here, we first demonstrated that KLHL8 targets ZAR1 and regulates its ubiquitination-induced degradation, which is a key step in oocyte maternal mRNA decay. Besides, oocyte maturation is accompanied by a transition from maternal mRNA stability to instability (Jiang et al, 2023), and insufficient maternal mRNA decay and abnormally high translation activity indirectly cause the spindle assembly checkpoint (SAC) activation and lead to prometaphase arrest of the oocyte cell cycle (Sha et al, 2018). The failure of maternal mRNA decay might be the final cause of maturation defects in *Klhl8°°−/−* oocytes. The relationship between ubiquitination and mRNA metabolism is linked through RNA binding proteins. ZAR1 regulates mRNA metabolism by modulating MARDO dissolution, while KLHL8 can regulate ZAR1 degradation through ubiquitination. Thus, KLHL8 indirectly affect mRNA metabolism by promoting the ubiquitination degradation of maternal RNA binding protein. This study thus suggests a novel role for the ubiquitin-proteasome pathway in both maternal protein and mRNA clearance during oocyte maturation.

Collectively, this study suggests KLHL8 was essential for female fertility by regulating the maternal protein homeostasis in mouse oocytes. We have established the interaction network between ubiquitination modification, RNA-binding proteins and maternal mRNA, and laid the foundation for elucidating the mechanisms of oocyte development and maturation.

## Methods

### Reagents and tools table

| Reagent/resource | Reference or source | Identifier or catalog number |
|---|---|---|
| **Experimental models** | | |
| HeLa cells (*H. sapiens*) | Cell Bank of the Shanghai Institute for Biological Sciences, the Chinese Academy of Sciences | SCSP-504 |
| C57BL6/J (*M. musculus*) | cyagen | N/A |
| C57BL6/J (*M. musculus*) | Charles river | N/A |
| **Recombinant DNA** | | |
| PCMV6-3′FLAG | This study | |

| Reagent/resource | Reference or source | Identifier or catalog number |
|---|---|---|
| PCMV6-3′HA | This study | |
| **Antibodies** | | |
| Rabbit anti-KLHL8 | Thermo Fisher | PA5-53640 |
| Rabbit anti-ZAR1 | Provided by Professor Hengyu Fan | |
| Rabbit anti-HA | Abclonal | AE036 |
| Mouse anti-FLAG | GNI | GNI4110-FG |
| Rabbit anti-tubulin | Abcam | ab11309 |
| Rabbit anti-TOM20 | Abclonal | A6774 |
| Rabbit anti-DDX6 | CST | 9407S |
| Rabbit anti-YBX2 | Abcam | ab154829 |
| Rabbit anti-DAZL | Abcam | ab215718 |
| Rabbit anti-K48-linkage Specific Polyubiquitin | CST | 8081S |
| Rabbit anti-Vinculin | CST | 13901 |
| Mouse anti-GAPDH | Abclonal | AC002 |
| Rabbit anti-LSM14B | Novus Biologicals | NBP2-56828 |
| Rabbit anti-ZFP36L2 | CST | 85891S |
| Rabbit anti-Cyclin B1 | Proteintect | 55004-1-AP |
| Rabbit anti-CDK1 | CST | 77055 |
| Rabbit anti-P-CDK1 | CST | 9111 |
| Rabbit anti-G3BP2 | Proteintect | 16276-1-AP |
| Rabbit anti-ZAR2 | Abmart | PC13550 |
| Goat anti-rabbit HRP | Abmart | M21002 |
| Goat anti-mouse HRP | Abmart | M21001 |
| **Oligonucleotides and other sequence-based reagents** | | |
| PCR primers | This study | Appendix Table S1 |
| Oligo (dT) | This study | Appendix Table S1 |
| **Chemicals, enzymes and other reagents** | | |
| M2 medium | Aibei Biotechnology | M1205 |
| PMSG | Aibei Biotechnology | M2620 |
| hCG | Aibei Biotechnology | M2520 |
| Milrinone | MCE | HY-14252 |

| Reagent/resource | Reference or source | Identifier or catalog number |
|---|---|---|
| Hyaluronidase | Aibei Biotechnology | M2215 |
| DMEM high glucose | meilun | MA0212 |
| Fetal bovine serum (FBS) | Gibco | A5256701 |
| Penicillin–streptomycin | meilun | MA0110 |
| 0.25% Trypsin | meilun | MA0233 |
| Poly Jet In Vitro DNA Transfection Reagent | Signa Gen | SL100688 |
| sodium dodecyl sulfate (SDS) | WEIAO Biotechnology | WB0133 |
| SDS-PAGE gels | WSHT Biotechnology | GSH2001 |
| Nitrocellulose membranes | PALL Life Sciences | 66485 |
| PBS | meilun | MA0015 |
| Tween-20 | SINOPHARM | 30189328 |
| Bovine serum albumin (BSA) | Sigma | B2064 |
| Nonfat milk | Sangon | A600669 |
| Protease inhibitor cocktail | Selleck | B14001 |
| FLAG-beads | Selleck | B23102 |
| HA-beads | Selleck | B26202 |
| MG132 | Selleck | S2619 |
| Triton X-100 | Sigma | T8787 |
| Hoechst | Thermo Fisher Scientific | R37605 |
| MitoTracker Red | Beyotime | C1035 |
| TMRE | MCE | HY-D0985A |
| NP40 | Sangon | A600385 |
| Paraformaldehyde | Sigma | FX0415 |
| Dextran sulfate sodium salt | Sangon | A600160 |
| RNase inhibitor | New England Biolabs | M0314S |
| Saline-Sodium Citrate buffer (SSC) | Sangon | B548109 |
| Formamide | Sangon | A600212 |
| ECL Western Blotting Substrate | Tanon | 180-5001 |
| **Software** | | |
| GraphPad Prism | GraphPad Software Inc. | Version 9.4 |
| ImageJ (Fiji) | github.com/fiji/fiji | |
| Adobe Illustrator | Adobe | Version 2024 |
| ZEISS Zen Lite | ZEISS | |
| **Other** | | |
| REPLI-g WTA Single Cell Kit | QIAGEN | 150063 |
| RNeasy Plus Micro Kit | QIAGEN | 74034 |
| PrimeScript RT reagent Kit with gDNA Eraser | Takara | RR047A |

| Reagent/resource | Reference or source | Identifier or catalog number |
|---|---|---|
| TB Green Premix Ex Taq | TaKaRa | RR820A |
| cDNA synthesis kit | Abcam | G490 |
| Hieff Clone One-Step Cloning Kit | Yeasen | 10905ES25 |
| KOD-Plus Mutagenesis Kit | TOYOBO | SMK-101 |
| HiScribe T7 ARCA mRNA Kit | New England Biolabs | E2060S |
| ROS detection assay kit | Beyotime | S0033S |
| ATP Testing Assay Kit | Thermo Fisher | A22066 |
| Single Cell/Low Input RNA Library Prep Kit | New England Biolabs | E6420 |
| Zeiss LSM 880 | ZEISS | N/A |
| Chemiluminescent imaging system | Tanon | 5200 |

## Ethics approval

All mouse experiments were reviewed and approved by the Shanghai Medical College of Fudan University (202205004S).

## Generation of mouse lines

All mice were maintained in a specific pathogen-free environment. $Klhl8^{fl/fl}$, $Zp3$-$Cre$ male mice were generated by crossing $Klhl8^{fl/+}$ females with Tg ($Zp3$-$Cre$) males in the C57BL/6J genetic background. Experimental mice were generated from homozygous-floxed females with homozygous-floxed males positive for $Zp3$-$Cre$. A total of 2593 bp of the $Klhl8$ gene was deleted, which caused complete removal of exons 3–4. $Klhl8^{fl/fl}$, $Zp3$-$Cre$ mice are abbreviated as $Klhl8^{oo-/-}$ mice in the article. For the identification of $Klhl8^{oo-/-}$ mice, DNA was extracted and the target fragment was amplified followed by sequence analysis. The primers are shown in Appendix Table S1.

## Breeding trials

Mice were maintained in temperature and humidity-controlled condition with food and water provided ad libitum and on 12-h light–dark cycle. They were housed in group of 4 mice/cage. All cages were placed in specific pathogen-free (SPF) conditions during the experiment. Fertility tests were carried out by crossing experimental females (6–8 weeks old) with WT C57BL/6J males. Female mice were checked for vaginal plugs every morning. Females were mated continuously for at least 6 months. The number of pups on the first day after parturition was counted as the litter size.

## Collection of mouse oocytes

For collection of GV oocytes, mice were humanely euthanized and ovaries were dissected from 8-week-old female mice 46 h after injection with 10 IU of pregnant mare serum gonadotrophin (PMSG).

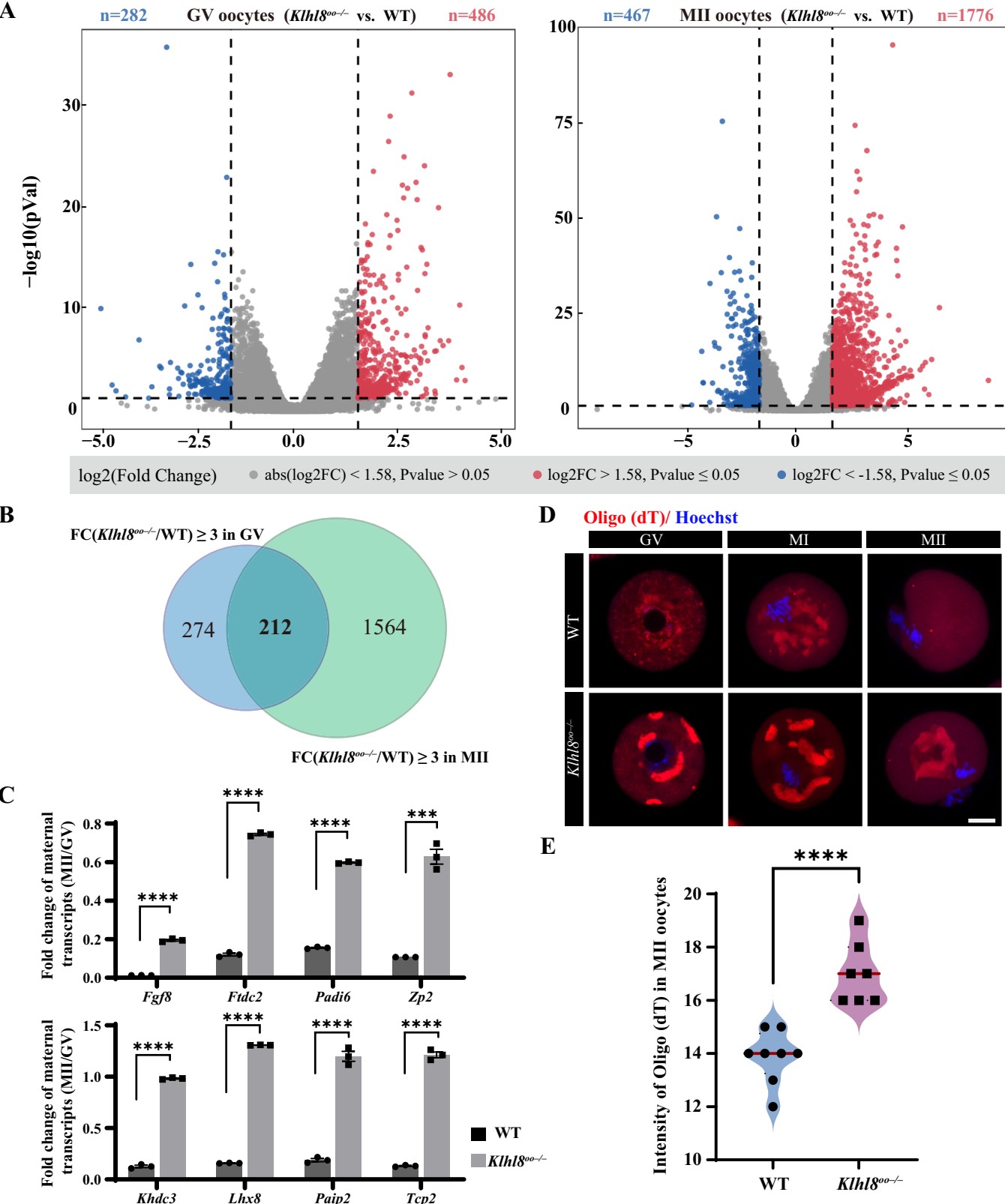

**A**

n=282 **GV oocytes (*Klhl8^{oo−/−}* vs. WT)** n=486

n=467 **MII oocytes (*Klhl8^{oo−/−}* vs. WT)** n=1776

−log10(pVal)

log2(Fold Change)

● abs(log2FC) < 1.58, Pvalue > 0.05 ● log2FC > 1.58, Pvalue ≤ 0.05 ● log2FC < −1.58, Pvalue ≤ 0.05

**B**

FC(*Klhl8^{oo−/−}*/WT) ≥ 3 in GV

274 **212** 1564

FC(*Klhl8^{oo−/−}*/WT) ≥ 3 in MII

**C**

Fold change of maternal transcripts (MII/GV)

**** **** **** ***

*Fgf8* *Ftdc2* *Padi6* *Zp2*

Fold change of maternal transcripts (MII/GV)

**** **** **** ****

*Khdc3* *Lhx8* *Paip2* *Tcp2*

■ WT ■ *Klhl8^{oo−/−}*

**D**

Oligo (dT)/ Hoechst

GV MI MII

WT

*Klhl8^{oo−/−}*

**E**

Intensity of Oligo (dT) in MII oocytes

****

WT *Klhl8^{oo−/−}*

**Figure 7. Maternal mRNA clearance is defective in *Klhl8*$^{oo-/-}$ oocytes.**

(A) Volcano plot comparing the transcripts of oocytes from WT and *Klhl8*$^{oo-/-}$ mice (GV and MII stage oocytes). Transcripts that were upregulated or downregulated by >threefold in *Klhl8*$^{oo-/-}$ oocytes are highlighted in red or blue, respectively. (B) Venn diagrams showing the overlap in transcripts that were upregulated (*Klhl8*$^{oo-/-}$ / WT > 3-fold) at the GV and MII stages. *P* values were calculated using Wald test. (C) qPCR results showing the fold changes of selected maternal transcripts in oocytes at the MII stage from WT and *Klhl8*$^{oo-/-}$ mice. Data are presented as mean ± SEM (*n* = 3). Statistical significance was determined using an unpaired two-tailed *t* test: ****$P$ < 0.0001, ***$P$ = 0.0002. (D) Representative images of poly(A) mRNA in GV, MI, and MII oocytes from WT and *Klhl8*$^{oo-/-}$ mice. Red, oligo(dT). Scale bar, 20 µm. (E) The quantification of oligo(dT) intensity in WT and *Klhl8*$^{oo-/-}$ MII oocytes. The average poly(A) mRNA intensity in oocytes was determined using ImageJ software (WT, *n* = 8; *Klhl8*$^{oo-/-}$, *n* = 7). Data are presented as mean ± SEM. Statistical significance was determined using an unpaired two-tailed *t* test: ****$P$ = 8 × 10$^{-5}$. Source data are available online for this figure.

Prophase I-arrested GV-stage oocytes were isolated by physical dissection of ovaries in M2 medium with 2.5 µM milrinone. MII oocytes were obtained at 14 h after release from milrinone.

For the collection of MII oocytes in vivo, 8-week-old female mice were injected with 10 IU PMSG. After 46 h the mice were injected with 10 IU of hCG, and 13 h later then MII-stage oocytes were collected. Cumulus–oocyte complexes (COCs) were harvested from the oviducts and were digested with hyaluronidase (300 IU/ml) to remove the cumulus cells.

## RNA extraction and qRT-PCR validation

A REPLI-g WTA Single Cell Kit (QIAGEN, 150063) was used for extraction of total RNA from oocytes, and cDNA was obtained according to the manufacturer's protocols. A RNeasy Plus Micro Kit (QIAGEN, 74034) was used for extraction of total RNA from tissues. Genomic DNA was removed and RNA reverse transcription was performed to obtain cDNA using a PrimeScript RT reagent Kit with gDNA Eraser (Takara, RR047A). The real-time quantitative reverse transcription PCR (qRT-PCR) was performed using a Quant Studio 6 Flex real-time system with TaKaRa TB Green Premix Ex Taq (Tli RNaseH Plus). The PCR reactions were performed as follows: 95 °C for 2 min and 40 cycles of 95 °C for 1 s and 60 °C for 30 s. Relative mRNA levels were measured in triplicate and normalized to the level of endogenous β-actin mRNA (internal control). The relative transcription levels of the samples were compared with those of the controls, followed by subsequent determination of the fold change. Finally, the relative expression levels of targeted genes were calculated using the $2^{-\Delta\Delta CT}$ method. The primers are shown in Appendix Table S1.

## Expression constructs

Total RNA was extracted from mouse GV oocytes and was used to synthesize the first-strand cDNA using a cDNA synthesis kit (Abcam, G490). The KLHL8 coding sequence region was amplified by PCR and cloned into the PCMV6 entry vector with an HA tag at the C-terminus using a Hieff Clone One-Step cloning kit (Yeasen, 10905ES25). The KLHL8, ZAR1 and YBX2 coding sequences were cloned into the PCMV6 entry vector with a FLAG tag at the C-terminus. Site-directed mutagenesis was performed with the KOD-Plus Mutagenesis Kit according to the manufacturer's instructions (TOYOBO, SMK-101).

## Cell culture and transfection

HeLa cells were obtained from the Cell Bank of the Shanghai Institute for Biological Sciences, the Chinese Academy of Sciences (Shanghai, China). Cell lines were recently authenticated and tested for mycoplasma contamination. Cells were cultured in Dulbecco's

modified Eagle's medium supplemented with 10% fetal bovine serum and 1% penicillin–streptomycin (Gibco, Waltham, MA, USA) in an atmosphere of 5% $CO_2$ at 37 °C to between 70 and 80% confluence. Plasmids were transfected into HeLa cells using the Poly Jet In Vitro DNA Transfection Reagent (Signa Gen) according to the manufacturer's instructions.

## Immunoblotting, co-immunoprecipitation, and ubiquitylation assay

Mouse oocytes (30 oocytes per sample) were collected and lysed in sodium dodecyl sulfate (SDS) lysis buffer, and all samples were heated at 100 °C for 10 min. Equal amounts of samples were electrophoresed on SDS-PAGE gels and transferred to nitrocellulose membranes. The membranes were blocked with 5% nonfat milk in PBS with 0.1% Tween-20 and then incubated with primary antibodies diluted in 5% bovine serum albumin (BSA) at 4 °C overnight. The primary antibodies were anti-KLHL8 (Thermo Fisher, PA5-53640, 1:1000 dilution), anti-ZAR1 (Kindly provided by professor Hengyu Fan, 1:2000 dilution), anti-YBX2 (Abcam, ab154829, 1:2000 dilution), anti-DDX6 (CST, 9407S, 1:2000 dilution), anti-DAZL (Abcam, ab215718, 1:1000 dilution), anti-ZFP36L2 (CST, 85891S, 1:1000 dilution), anti-LSM14B (Novus Biologicals, NBP2-56828, 1:1000 dilution), anti-G3BP2 (Proteintect, 16276-1-AP, 1:1000 dilution), anti-ZAR2 (Abmart, PC13550, 1:1000 dilution), anti-CDK1 (CST, 77055, 1:1000 dilution), anti-P-CDK1 (CST, 9111, 1:1000 dilution), anti-Cyclin B1 (Proteintect, 55004-1-AP, 1:1000 dilution), anti-GAPDH (Abclonal, AC002, 1:3000 dilution), and anti-vinculin (CST, 13901, 1:2000 dilution). The membranes were visualized using ECL Western Blotting Substrate (Tanon) after incubation with secondary antibody for 1 h. The secondary antibodies were HRP-conjugated goat anti-rabbit IgG (Abmart, M21002, 1:5000 dilution) and goat anti-mouse IgG (Abmart, M21001, 1:5000 dilution). Signals were captured on a Tanon 5200 Imaging Workstation.

The KLHL8-HA plasmid was co-transfected with the ZAR1-FLAG and YBX2-FLAG plasmids into HeLa cells. At 36 h after transfection, the transfected HeLa cells were quickly washed in cold PBS twice and lysed in NP40 lysis buffer with 1% protease inhibitor cocktail (Bimake, Houston, TX, USA) at 4 °C for 30 min, and then centrifuged at 12,000 × *g* for 30 min to harvest the supernatant. FLAG-beads or HA-beads were added to each sample and incubated at 4 °C for 2 h, and the beads were washed with lysis buffer four times. The bead-bound proteins were eluted using SDS sample buffer, resolved by SDS-polyacrylamide gel electrophoresis (SDS-PAGE), transferred to nitrocellulose membranes (Pall Corporation), and probed with mouse anti-FLAG (GNI, GNI4110-FG, 1:3000 dilution) or rabbit anti-HA (Abclonal, AE036, 1:3000 dilution) antibodies. The secondary antibodies were goat anti-rabbit IgG (Abmart, 1:5000 dilution) or goat anti-mouse IgG (Abmart, 1:5000 dilution) conjugated to horseradish peroxidase.

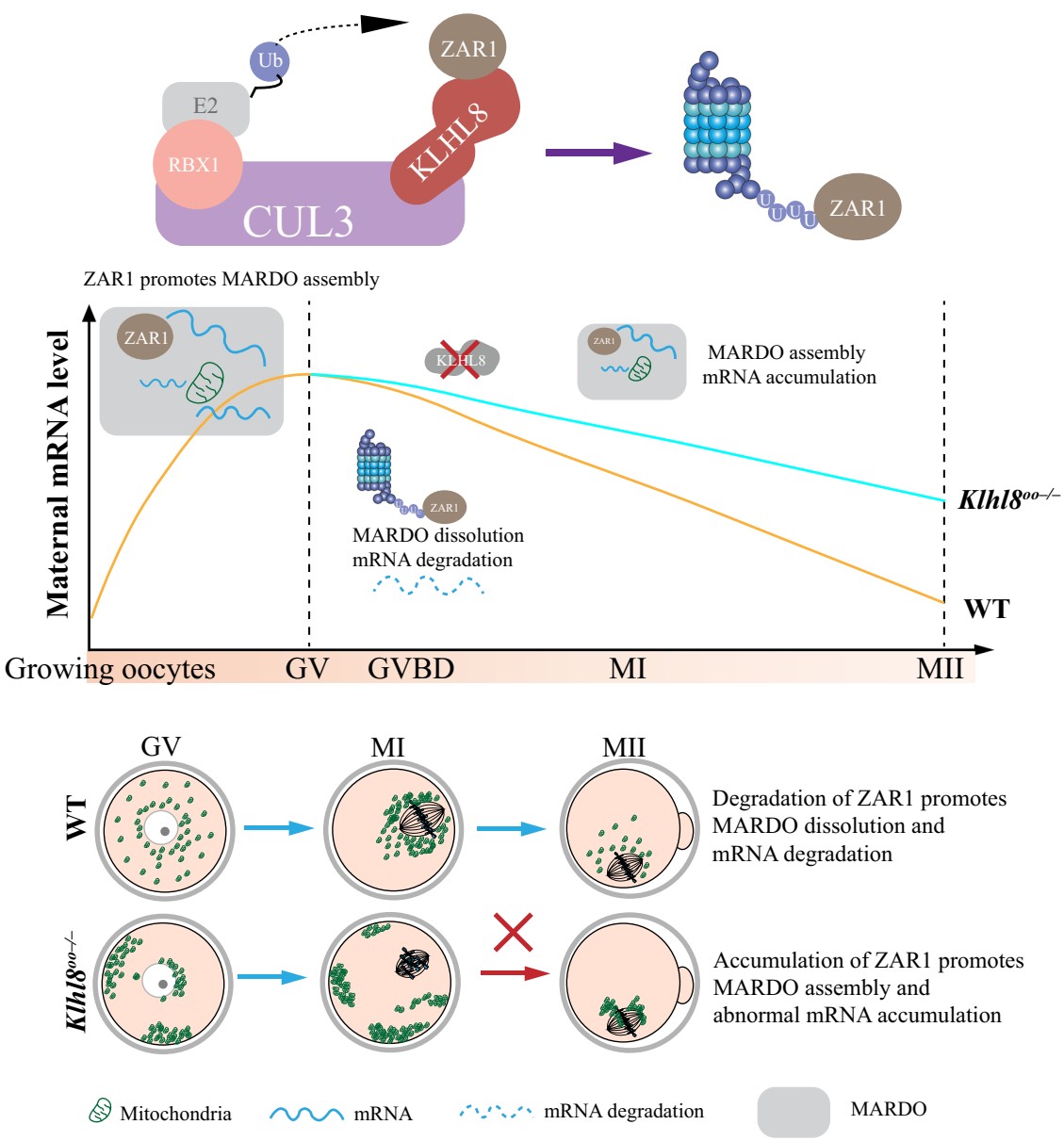

**Figure 8. Diagram for KLHL8 targets ZAR1 to regulate mRNA clearance in oocytes.**

KLHL8-mediated ubiquitination and degradation of ZAR1 regulates the dissolution of the MARDO and mRNA clearance during oocyte meiotic maturation.

## mRNA preparation and microinjection

To prepare mRNA for microinjection, the linearized fusion plasmid was used as the template for production of capped mRNA using the HiScribe T7 ARCA mRNA Kit (with tailing) (New England Biolabs, E2060S). The GV oocytes from WT and *Klhl8^oo−/−* female mice were cultured in M2 medium with 2.5 µM milrinone after *Klhl8* mRNA microinjection (1000 ng/µL). Metaphase II oocytes were obtained at 14 h after release from milrinone. The phenotype of the *Klhl8^oo−/−* mouse model was rescued by injection of *Klhl8* mRNA. WT and *Klhl8^oo−/−* oocytes were randomized to be injected with mRNA. Blinding was used for data analysis and in vitro microinjection.

## Time-lapse live-cell imaging

WT and *Klhl8^oo−/−* GV oocytes were injected with 10 pL of 1000 ng/µL GFP mRNA and cultured in M2 medium containing 2.5 µM milrinone for 2 h to allow GFP expression. The oocytes were then transferred to EVOS M7000 microscope (Invitrogen) for fluorescence imaging.

## Immunofluorescence

Oocytes were collected and fixed in 4% PBS-buffered formalin for 30 min and then permeabilized with 0.3% Triton X-100 in PBS for 1 h at room temperature. The oocytes were then blocked in 1% BSA-supplemented PBS for 1 h at room temperature followed by

incubation with primary antibodies at 4 °C overnight. Primary antibodies against FLAG (GNI, GNI4110-FG, 1:200 dilution), ZAR1 (Kindly provided by Professor Hengyu Fan, 1:200 dilution), YBX2 (Abcam, ab154829, 1:200 dilution), TOM20 (Abclonal, A6774, 1:200 dilution), and Cy3-beta Tubulin (Abcam, ab11309, 1:200 dilution) were used in this study. After washing three times, oocytes were incubated with the appropriate secondary antibody for 1 h at room temperature before staining the DNA for 10 min. Finally, oocytes were imaged with a Zeiss LSM 880 laser scanning confocal microscope.

## Visualization of the mitochondria by MitoTracker Red

MitoTracker Red (Beyotime, 1:1000 dilution) was first diluted with M2 medium to make culture droplets and placed in an incubator at 37 °C to equilibrate. Oocytes were then transferred to the droplets and incubated at 37 °C for 15 min. After counterstaining with Hoechst (Thermo Fisher Scientific, R37605, 1:700 dilution), the oocytes were moved to a glass bottom culture dish and imaged under a laser scanning confocal microscope. The distribution of mitochondria was reflected by MitoTracker Red staining. All photos for analysis were taken with the same intensity parameters and exposure time settings. The degree of clustering was assessed by calculating the reciprocal of the mitochondrial cluster number (Cheng et al, 2022).

## ROS detection

ROS levels were determined with a ROS detection assay kit (Beyotime) according to the manufacturer's instructions. Briefly, oocytes were stained in M2 medium with DCFH-DA for 30 min and washed in M2 medium three times, and the fluorescence intensity of ROS was examined under an EVOS M7000 Imaging System (Invitrogen).

## MMP measurement

The MMP was determined with TMRE (MCE, HY-D0985A). TMRE is a rapidly and selectively mitochondrial membrane potential fluorescent probe with red fluorescence. According to the manufacturer's instructions, oocytes were stained in M2 medium with TMRE (10 μM) for 10 min and washed in M2 medium three times. Then, the fluorescence intensity was measured under a Zeiss LSM 880 laser scanning confocal microscope.

## Luminescence testing for ATP quantification

The ATP content of oocytes was measured using the ATP Testing Assay Kit (Thermo Fisher, A22066) according to the manufacturer's instructions. Briefly, ten oocytes were lysed in lysis buffer. After being centrifuged at $12,000 \times g$ for 10 min, the samples were mixed with testing buffer, and the ATP concentrations were measured on a luminescence detector.

## Proteomics data analysis

A total of 25 GV oocytes were collected from WT and *Klhl8*$^{oo−/−}$ mice with three replicates per sample. After washing three times in PBS, the samples were collected in tubes with lysis buffer and then transferred to a −80 °C freezer in liquid nitrogen. The samples were sent on dry ice to Shanghai EasyDIA Biotechnology Co., Ltd, which was responsible for proteomics testing and data analysis. A cut-off of the adjusted *P* value of 0.05 (false discovery rate adjusted) along with a fold change of 1.4 was applied to determine significantly regulated proteins in the WT and *Klhl8*$^{oo−/−}$ comparison. Proteomic data are shown in Dataset EV1.

## RNA-seq and data analysis

A total of 20 oocytes at each of the two stages (GV and MII) were collected from WT and *Klhl8*$^{oo−/−}$ mice, with four replicates per sample. After washing three times in PBS, the samples were collected in tubes with lysis component and RNase inhibitor, and full-length cDNAs were generated using a Single Cell/Low Input RNA Library Prep Kit (New England Biolabs, E6420). The samples were sent on dry ice to Genergy Biotech (Shanghai) Co., Ltd., which was responsible for transcriptome sequencing and data analysis. The FPKMs of the RNA-seq results are listed in Dataset EV2.

## mRNA-FISH

Oocytes were fixed in 2% paraformaldehyde at room temperature for 30 min and permeabilized in 0.1% Triton-X100 in PBS with RNase inhibitor (New England Biolabs, M0314S) for 1 h. Oocytes were washed in prehybridization buffer (10% 20× Saline-Sodium Citrate buffer (SSC) and 10% formamide) and incubated for 16 h at 37 °C in hybridization buffer (10% w/v dextran sulfate in prehybridization buffer) with oligo (dT) probe. Oocytes were then washed three times in washing buffer (2× SSC with 0.1% Tween-20) for 5 min. Hoechst staining was used for visualization of chromatin structures. The oocytes were scanned with an LSM 880 laser scanning confocal microscope (Zeiss).

## Statistical analysis

All data were from at least two independent experiments. Quantitation of western blotting results was performed with ImageJ software, and statistical analyses were performed using GraphPad Prism. The method of statistical test and *P* values were shown in the figure legends. The data are presented as mean ± SEM or mean ± SD. *P* values are indicated by asterisks in the figures.

# Data availability

RNA-seq raw sequence data reported in this paper have been deposited in the Genome Sequence Archive in National Genomics Data Center, China National Center for Bioinformation/Beijing Institute of Genomics, Chinese Academy of Sciences (GSA: CRA021060) that are publicly accessible at https://ngdc.cncb.ac.cn/gsa. The raw proteomic data reported in this paper have been deposited in the OMIX, China National Center for Bioinformation / Beijing Institute of Genomics, Chinese Academy of Sciences (https://ngdc.cncb.ac.cn/omix: accession no. OMIX010161) that are publicly accessible at https://ngdc.cncb.ac.cn/omix/releaseList.

The source data of this paper are collected in the following database record: biostudies:S-SCDT-10_1038-S44319-025-00537-y.

## Peer review information

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

## Acknowledgements

This work was supported by the National Key Research and Development Program for Young Scientists (2022YFC2702300), the National Natural Science Foundation of China (82325021, 82288102, 32130029, 82371662), the National Key Research and Development Program of China (2021YFC2700100, 2024YFC2706600), the Fund of Fudan University and Cao'ejiang Basic Research (24FCB01), the New Cornerstone Science Foundation through the XPLORER PRIZE. Lei Wang is a SANS Exploration Scholar. We thank Professor Hengyu Fan (Zhejiang University) for kindly providing ZAR1 antibody.

## Author contributions

**Huizhen Fan**: Data curation; Formal analysis; Validation; Investigation; Visualization; Methodology; Writing—original draft; Writing—review and editing. **Ruyi Liu**: Resources; Methodology. **Ran Yu**: Resources; Methodology. **Biaobang Chen**: Data curation; Formal analysis. **Qiaoli Li**: Resources. **Jian Mu**: Resources; Methodology. **Weijie Wang**: Resources; Methodology. **Tianyu Wu**: Resources; Methodology. **Lin He**: Resources. **Lei Wang**: Conceptualization; Resources; Funding acquisition; Project administration. **Qing Sang**: Conceptualization; Supervision; Funding acquisition; Project administration; Writing—review and editing. **Zhihua Zhang**: Supervision; Funding acquisition; Project administration; Writing—review and editing.

Source data underlying figure panels in this paper may have individual authorship assigned. Where available, figure panel/source data authorship is listed in the following database record: biostudies:S-SCDT-10_1038-S44319-025-00537-y.

## Disclosure and competing interests statement

The authors declare no competing interests.

# Expanded View Figures

**A**

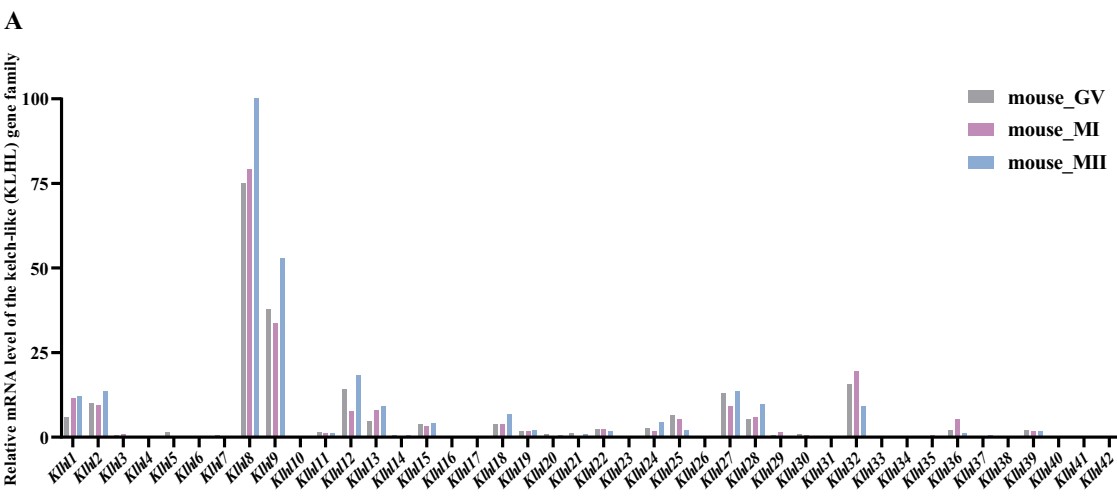

**B**

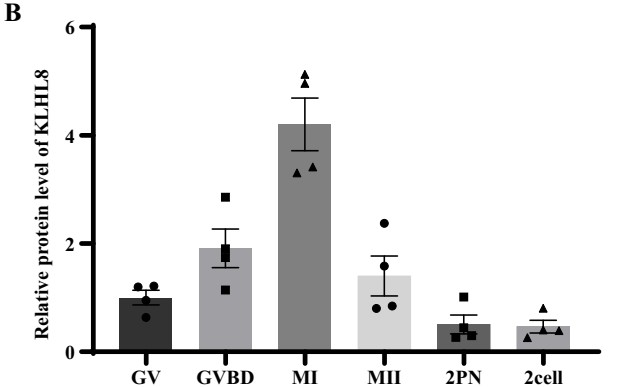

**Figure EV1. The transcription level and protein abundance of KLHL8 during oocyte maturation.**

(A) In-house database qRT-PCR results for KLHL family mRNA expression in GV, MI, and MII oocytes. (B) Quantification of KLHL8 protein abundance during oocyte maturation. Data are presented as mean ± SEM ($n = 4$).

A

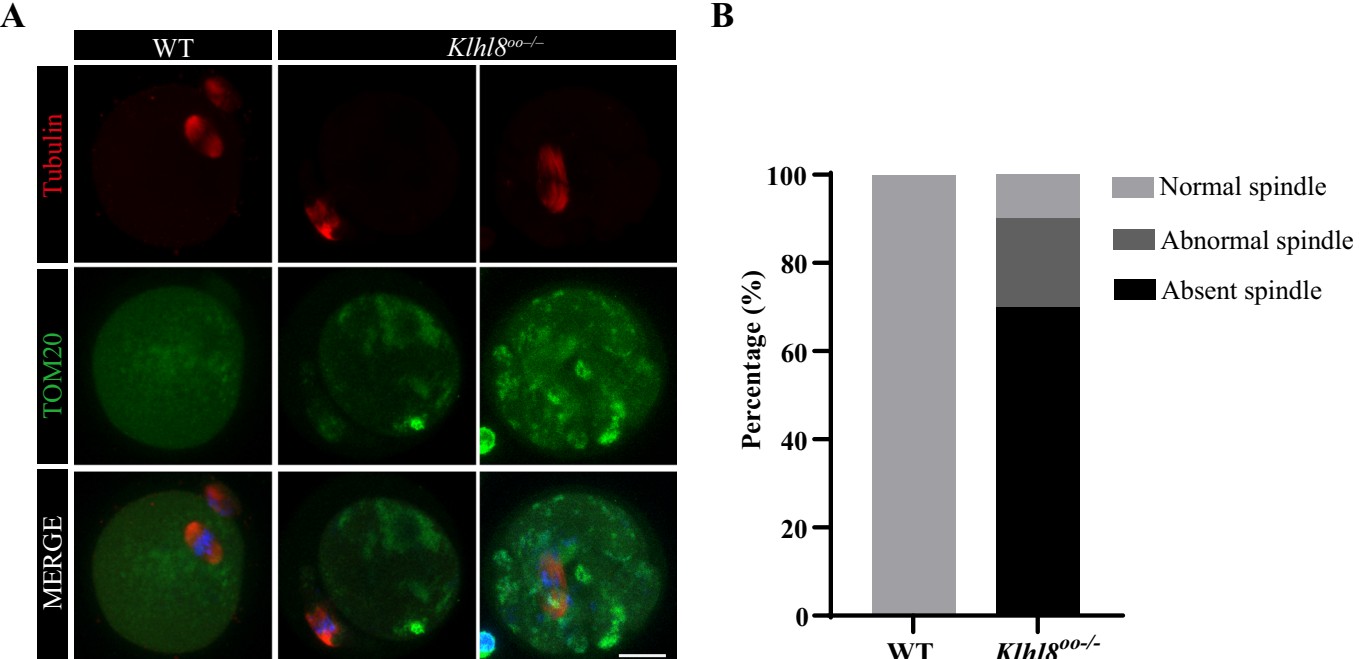

**Figure EV2.** *Klhl8* **deletion causes abnormal spindle in MII oocytes.**

(A) The spindle morphology in few oocytes that manage to extrude PB1 in vivo. Red, β-Tubulin; Green, mitochondria (TOM20). Scale bar, 20 μm. (B) The percentage of oocytes with different spindle morphologies in WT and *Klhl8oo−/−* MII oocytes (WT, *n* = 8; *Klhl8oo−/−*, *n* = 10).

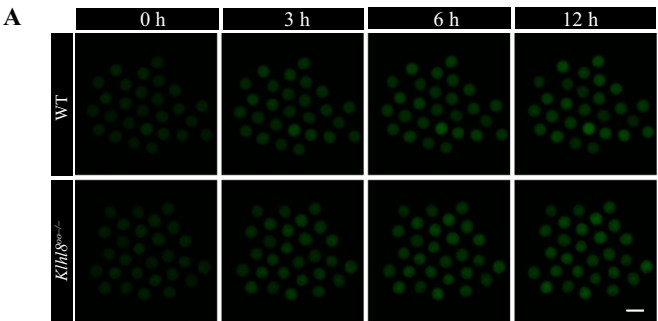

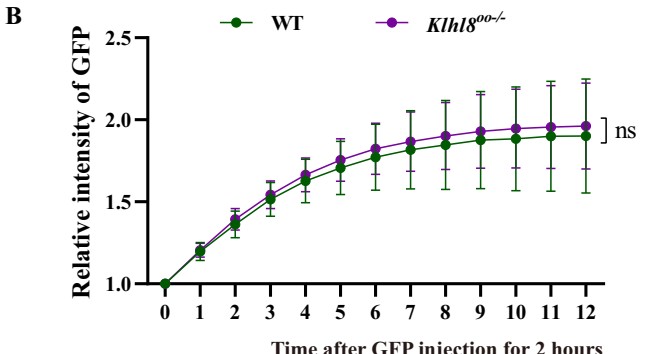

**Figure EV3.  Translation efficiency in WT and *Klhl8*ᵒᵒ⁻/⁻ oocytes.**

(**A**) GFP expression in WT and *Klhl8*ᵒᵒ⁻/⁻ GV oocytes after mRNA injection for 2 h. Scale bar, 100 μm. (**B**) The quantification of GFP intensity in WT and *Klhl8*ᵒᵒ⁻/⁻ GV oocytes (WT, $n = 28$; *Klhl8*ᵒᵒ⁻/⁻, $n = 29$). Data are presented as mean ± SD. Statistical significance was determined using Multiple unpaired *t* test: ns not significant.

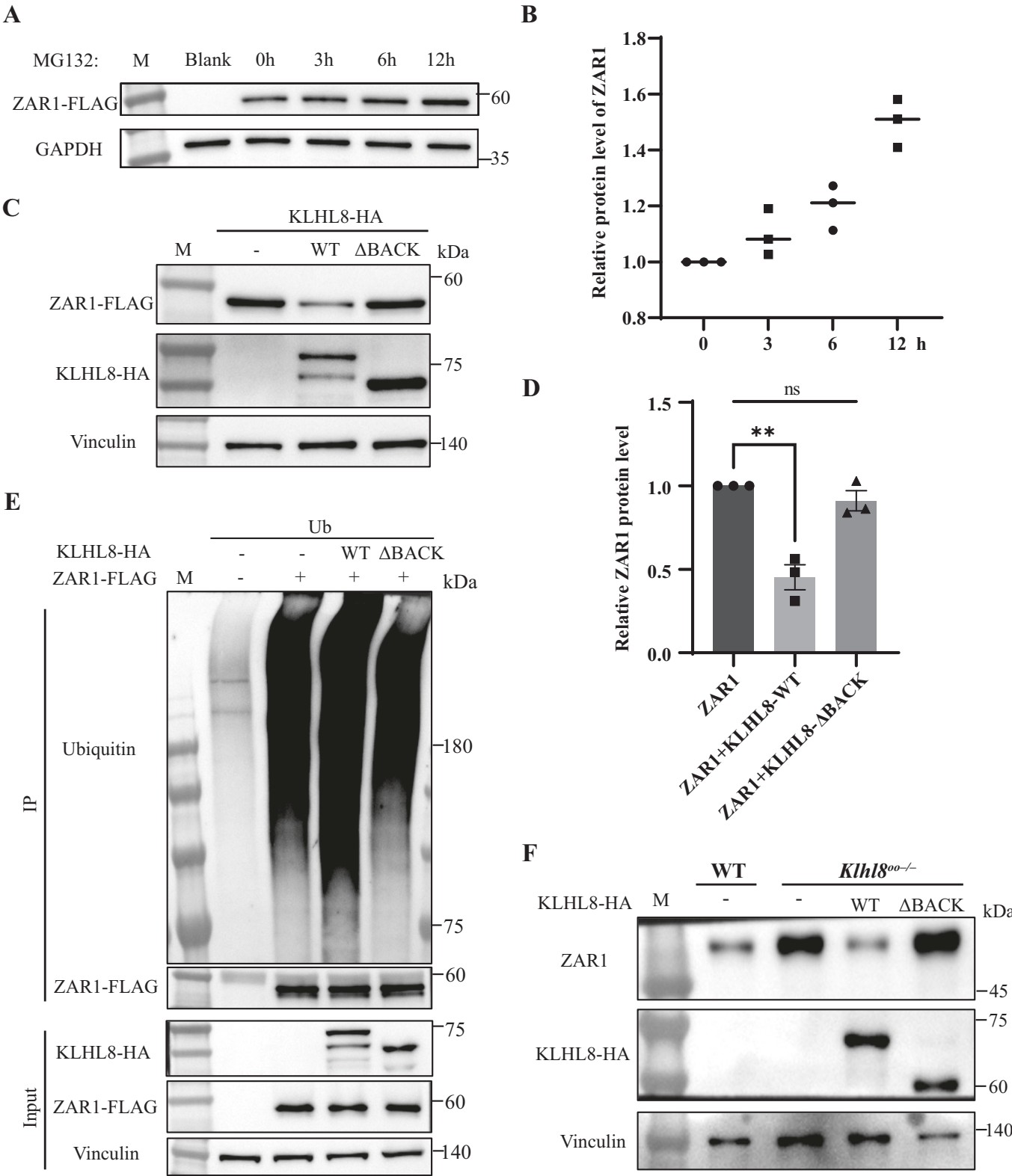

◀ **Figure EV4. BACK domain of KLHL8 is necessary for ZAR1 ubiquitination and proteasome degradation.**

(A) Western blot result showing ZAR1 protein level after adding MG132. (B) Quantification of the ZAR1 protein level after adding MG132 for different duration. Data are presented as mean ± SEM (n = 3). (C) Immunoblotting for ZAR1 with *Klhl8*-WT or *Klhl8*-ΔBACK overexpression in cultured HeLa cells. (D) Quantitative analysis of ZAR1 protein level with *Klhl8*-WT or *Klhl8*-ΔBACK overexpression in cultured HeLa cells. Data are presented as mean ± SEM (n = 3). Statistical significance was determined using an unpaired two-tailed *t* test: ns not significant, **P = 0.0018. (E) The ubiquitination of ZAR1 with *Klhl8*-WT or *Klhl8*-ΔBACK overexpression in cultured HeLa cells. (F) Immunoblotting for ZAR1 in WT and *Klhl8*^∞−/− GV oocytes with *Klhl8*-WT or *Klhl8*-ΔBACK mRNA injection. Vinculin was used as the protein loading control.

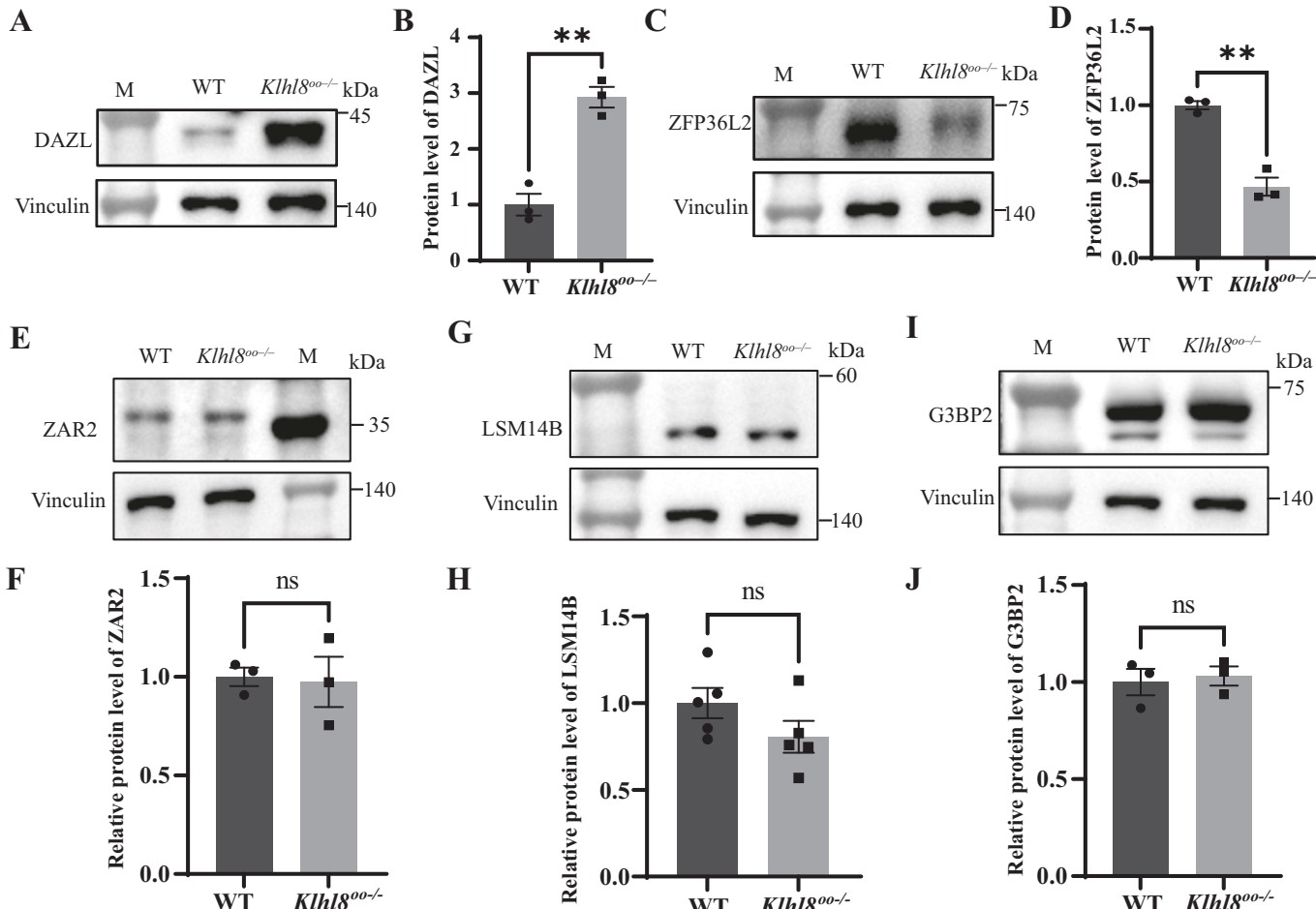

**Figure EV5. Effects of *Klhl8* deletion on other RNA binding proteins.**

(A, B) Western blot result and quantitative analysis of DAZL protein level in WT and *Klhl8^oo−/−* oocytes. Vinculin was used as the protein loading control. Data are presented as mean ± SEM (*n* = 3). Statistical significance was determined using an unpaired two-tailed *t* test: **P* = 0.002. (C, D) Western blot result and quantitative analysis of ZFP36L2 protein level in WT and *Klhl8^oo−/−* oocytes. Vinculin was used as the protein loading control. Data are presented as mean ± SEM (*n* = 3). Statistical significance was determined using an unpaired two-tailed *t* test: **P* = 0.0012. (E, F) Western blot result and quantitative analysis of ZAR2 protein level in WT and *Klhl8^oo−/−* oocytes. Vinculin was used as the protein loading control, *n* = 3. (G, H) Western blot result and quantitative analysis of LSM14B protein level in WT and *Klhl8^oo−/−* oocytes. Vinculin was used as the protein loading control, *n* = 5. (I, J) Western blot result and quantitative analysis of G3BP2 protein level in WT and *Klhl8^oo−/−* oocytes. Vinculin was used as the protein loading control, *n* = 3. In (F, H, J), data are presented as mean ± SEM. Statistical significance was determined using an unpaired two-tailed *t* test: ns not significant.

