## [Peer Review File · EMBO Reports]

The E3 ubiquitin ligase adaptor KLHL8 targets ZAR1 to regulate maternal mRNA degradation in oocytes

Huizhen Fan, Ruyi Liu, Ran Yu, Biaobang Chen, Qiaoli Li, Jian Mu, Weijie Wang, Tianyu Wu, Lin He, Lei Wang, Qing Sang, and Zihua Zhang

Corresponding author(s): Zihua Zhang (zihuazhang@fudan.edu.cn), Lei Wang (wangleiwanglei@fudan.edu.cn), Qing Sang (sangqing@fudan.edu.cn)

Review Timeline:

Submission Date:	18th Dec 24
Editorial Decision:	16th Jan 25
Revision Received:	29th Apr 25
Editorial Decision:	26th Jun 25
Revision Received:	2nd Jul 25
Accepted:	8th Jul 25

Editor: *Martina Rembold*

Transaction Report:

Dear Dr. Zhang

Thank you for the submission of your research manuscript to our journal. We have now received the full set of referee reports that is copied below.

As you will see, the referees acknowledge that the findings are interesting and that the conclusions are overall supported by the data presented but they also raise a number of concerns and have suggestions how to further strengthen the data. I agree with all the points and concerns raised by the referees and these need to be addressed in full. It will be important to strengthen the data on ZAR1 as being a functionally relevant target downstream of KLHL8 and to at least discuss that other potential targets might be involved in the oocyte and infertility phenotype as well. It is also important to test whether KLHL8 really degrades not only ubiquitinates ZAR1, also in the oocyte.

Given these constructive comments, we would like to invite you to revise your manuscript with the understanding that the referee concerns (as detailed above and in their reports) must be fully addressed and their suggestions taken on board. Please address all referee concerns in a complete point-by-point response. Acceptance of the manuscript will depend on a positive outcome of a second round of review. It is EMBO Reports policy to allow a single round of revision only and acceptance or rejection of the manuscript will therefore depend on the completeness of your responses included in the next, final version of the manuscript.

We realize that it is difficult to revise to a specific deadline. In the interest of protecting the conceptual advance provided by the work, we recommend a revision within 3 months (October 1st). Please discuss the revision progress ahead of this time with the editor if you require more time to complete the revisions.

I am also happy to discuss the exact revision requirements further via e-mail or a video call, if you wish.

=====
IMPORTANT NOTE:

We perform an initial quality control of all revised manuscripts before re-review. Your manuscript will FAIL this control and the handling will be delayed IN CASE the following APPLIES:

- 1) A data availability section providing access to data deposited in public databases is missing. If you have not deposited any data, please add a sentence to the data availability section that explains that.
- 2) Your manuscript contains statistics and error bars based on $n=2$. Please use scatter blots in these cases. No statistics should be calculated if $n=2$.

=====

- 1) a .docx formatted version of the manuscript text (including legends for main figures, EV figures and tables). Please make sure that the changes are highlighted to be clearly visible.
- 2) individual production quality figure files as .eps, .tif, .jpg (one file per figure). Please download our Figure Preparation Guidelines (figure preparation pdf) from our Author Guidelines pages <https://www.embopress.org/page/journal/14693178/authorguide> for more info on how to prepare your figures.
- 3) a .docx formatted letter INCLUDING the reviewers' reports and your detailed point-by-point responses to their comments. As part of the EMBO Press transparent editorial process, the point-by-point response is part of the Review Process File (RPF), which will be published alongside your paper.
- 4) a complete author checklist, which you can download from our author guidelines (<<https://www.embopress.org/page/journal/14693178/authorguide>>). Please insert information in the checklist that is also reflected in the manuscript. The completed author checklist will also be part of the RPF.
- 5) Please note that all corresponding authors are required to supply an ORCID ID for their name upon submission of a revised manuscript (<<https://orcid.org/>>). Please find instructions on how to link your ORCID ID to your account in our manuscript

tracking system in our Author guidelines

(<<https://www.embopress.org/page/journal/14693178/authorguide#authorshipguidelines>>)

6) We replaced Supplementary Information with Expanded View (EV) Figures and Tables that are collapsible/expandable online. A maximum of 5 EV Figures can be typeset. EV Figures should be cited as 'Figure EV1, Figure EV2' etc... in the text and their respective legends should be included in the main text after the legends of regular figures.

7) Before submitting your revision, primary datasets (and computer code, where appropriate) produced in this study need to be deposited in an appropriate public database (see <<https://www.embopress.org/page/journal/14693178/authorguide#dataavailability>>). Specifically, we would kindly ask you to provide public access to the RNA-seq and proteomics datasets.

The accession numbers and database should be listed in a formal "Data Availability " section (placed after Materials & Method) that follows the model below (see also <<https://www.embopress.org/page/journal/14693178/authorguide#dataavailability>>). Please note that the Data Availability Section is restricted to new primary data that are part of this study.

Data availability

Additional information on source data and instruction on how to label the files are available <<https://www.embopress.org/page/journal/14693178/authorguide#sourcedata>>.

10) Figure legends and data quantification:

- the name of the statistical test used to generate error bars and P values,
- the number (n) of independent experiments (please specify technical or biological replicates) underlying each data point,
- the nature of the bars and error bars (s.d., s.e.m.)
- If the data are obtained from n {less than or equal to} 5, show the individual data points in addition to the SD or SEM.
- If the data are obtained from n {less than or equal to} 2, use scatter blots showing the individual data points.

11) Our journal encourages inclusion of *data citations in the reference list* to directly cite datasets that were re-used and obtained from public databases. Data citations in the article text are distinct from normal bibliographical citations and should directly link to the database records from which the data can be accessed. In the main text, data citations are formatted as follows: "Data ref: Smith et al, 2001" or "Data ref: NCBI Sequence Read Archive PRJNA342805, 2017". In the Reference list, data citations must be labeled with "[DATASET]". A data reference must provide the database name, accession number/identifiers and a resolvable link to the landing page from which the data can be accessed at the end of the reference. Further instructions are available at <<https://www.embopress.org/page/journal/14693178/authorguide#referencesformat>>.

12) All Materials and Methods need to be described in the main text using our 'Structured Methods' format. According to this format, the Methods section includes a Reagents and Tools Table (listing key reagents, experimental models, software and relevant equipment and including their sources and relevant identifiers) followed by a Methods and Protocols section describing the methods, ideally using a step-by-step protocol format. The aim is to facilitate adoption of the methodologies across labs. Please download and fill our Reagents and Tools Table template (.docx), which you can find in our author guidelines:

13) As part of the EMBO publication's Transparent Editorial Process, EMBO Reports publishes online a Review Process File to accompany accepted manuscripts. This File will be published in conjunction with your paper and will include the referee reports, your point-by-point response and all pertinent correspondence relating to the manuscript.

Yours sincerely,

=====

Referee #1:

Despite the recent progresses in clarifying maternal transcript homeostasis during oocyte maturation and maternal-to-zygotic transition in mammals, the equally important maternal protein homeostasis, which is essential for meiotic cell-cycle progression and subsequent embryonic development, remain to be investigated in depth. In this study, the authors identified a new maternally enriched E3 ubiquitin ligase, KLHL8, is highly expressed in mouse oocytes and co-localizes with mitochondria. The functional importance of this maternal protein was demonstrated by the fact that oocyte-specific deletion of Klhl8 causes female infertility due to oocyte maturation defects. The results of mechanistic studies showed that ZAR1, an RNA binding protein which is required for mitochondria-associated ribonucleoprotein domain (MARDO) dissolution, is specifically recognized and degraded by KLHL8-mediated ubiquitination. In general, this manuscript is clearly written. The experiments were well designed, and the results appropriately presented. I have some suggestions and comments to potentially improve the current manuscript.

Major:

1. Author proved that KLHL8 target to ZAR1 and prevent the dissolution of MARDO, which caused the oocytes retardation in MI stage. However, according to the previous work (Cheng et.al, Science, 2022; Figure 4F), although the over-expression of mZar1 indeed delayed MARDO dissolution, the extrusion of PB1 was not significantly influenced. The results seem not consistent with the phenotype of Klhl8^{oo-/-} oocytes in Figure 2H. As there are many other differentially expressed proteins were detected in KLHL8 deleted oocytes, there may be other key substrates that cause the developmental retardation and infertility. More

discussion is needed about this.

2. Authors proved that over-expressing Klhl8 can decrease the defects in Klhl8^{oo-/-} oocytes, and declared the importance of KLHL8-ZAR1-MARDO pathway. However, the results is not sufficient to support the importance of ZAR1 degradation to the meiosis. Deletion of ZAR1 in Klhl8^{oo-/-} oocytes is needed.
3. According to Gabriele et.al (Cell, 2024), there is the other organelle containing proteasomal system in oocytes named ELVA that can selectively degrade protein aggregates during meiosis. As the mitochondrion are excluded from this structure, and ZAR1, LSM14B and G3BP2 had been confirmed not to be degraded by ELVAs, it would be interesting to discuss the relationship KLHL8-dependent ub-proteasomal system and ELVAs in oocytes. And the abundance of other two proteins (LSM14B and G3BP2) can also be detected in Klhl8^{oo-/-} oocytes.
4. Is ZAR2, or ZAR1L, the genetic and functional homolog of ZAR1 in mouse oocytes but was not being paid sufficient attentions as ZAR1, also a target of KLHL8 for degradation? This protein was not mentioned in the manuscript. But in my opinion, at least some descriptions of ZAR2 should be added in the Introduction, and its possibility as a KLHL8 target is needed to be discussed.

Minor:

1. In Figure 1B, the protein abundance of KLHL8 is specifically low in GVBD stage oocytes compared to GV or MI stage oocytes. Are there any physiological or technological explanations for this phenomena?
2. In Figure 2E and line 112, there are many abnormal cytoplasmic granules in Klhl8^{oo-/-} oocytes, what are the component of these granules?
3. In line 133-134, it was confirmed that the protein of ZAR1 is sharply increased in Klhl8^{oo-/-} oocytes with mRNA unchanged. Except the explanation that KLHL8 help with degrading ZAR1, the translation efficiency is also needed to be detected.
4. In line 173 and Figure 5C, what is 'mitochondrial cluster index'? Please explain.
5. In line 203-204, the relative RNA copy numbers normalized by spike-in or reference genes is needed to support the abnormal RNA abundance.
6. In Figure 4A, 'ZAR1-FALG' should be 'ZAR1-FLAG'.
7. In Figure 4C, the marker and the number of ZAR1-FLAG_input did not correspond, the same in Figure 4F_ZAR1-FLAG_IP, the band of ZAR1-FLAG_IP is too low. Please check.
8. In the figure legend of Figure 4F, authors declared that KLHL8 promoted the ubiquitination-mediated protein degradation of ZAR1. However, this figure just proved the KLHL8-dependent ubiquitination of ZAR1 but not the degradation. It needs to be more precise.
9. In Figure 5F, the JC-1-stained mitochondrion seemed to be circles, which is unusual. Please explain the mitochondrial morphology.
10. Some pertinent references should be cited during revision. For example, a review article comparing the homology and redundancy of ZAR1 and ZAR2 (PMID: 35072788), and a newly published study showing the association of ZAR1/2-mediated epigenetic regulations with oocyte aging (PMID: 39755931).

Referee #2:

The authors present an interesting manuscript on the role of the KLHL8 involved in the ubiquitination of proteins in the regulation of oocyte meiotic progression. They have shown that the absence of KLHL8 leads to female sterility due to the lack of extrusion of the polar body. The study is very interesting and contributes to the molecular physiology of oocytes. However, it needs to be significantly improved before publication.

Fig.1B Please provide quantitatively equal samples to describe expression of KLHL in oocytes. Based on Vinculin authors load diferent amount of cell in the stages.

Where is KLHL localized?

Fig.2EG Presentation of data not convincing. Provide, clear BF images of better quality with equal light intensity and contrast.
Fig.2GH Image and quantification doesn't positively correlates in term of PBE. Does cytoplasmic dimorphism refer to cytoplasm granulation? This morphology is common for C57BL/6J genetic background. Additional quantification of PNs appears as essential based on statement in Results section and Fig.2L. The absence of MII oocyte indicate MI arrest in which MPF plays a

key role. Does permanent activity of MPF in the KLHL^{-/-} oocytes leads to reported phenotype?

Fig.5F fluorescence signal is not visible.

As expected, the authors conclude that the stability of ZAR1 is regulated by UPS. However, the conclusion that KLHL is responsible for the degradation of ZAR1 was not confirmed in the oocyte.

Authors state that morphology of ER or Golgi is equal between genotypes (Fig.S6). Morphology/staining of ER or Golgi is unusual based what has been reported in literature.

In the Volcano plots Fig.7 highlight/point transcripts analyzed in this study. L207 Authors state that KLHL regulate transcriptome prior GV, please explain the conclusion by which data is supported. Additionally, due to absence of MII oocytes in the KLHL^{-/-} mice (Fig.2G) which groups were compared for DE here? Does RNA-seq and proteomic datasets correlate?

L256 Statement is not supported by the provided data and appears as speculative.

Discuss what has been presented in relation to ubiquitination and RNA metabolism.

What is the first band in the IB images?

Absence of number of cell analysed in experiments.

Referee #3:

The regulation of mRNA stability during oocyte maturation and early embryo development is essential to ensure proper embryonic development. In 2022, Cheng et al. identified a mitochondria-associated RNA domain (MARDO) in mammalian oocytes which is assembled by the oocyte-specific protein ZAR1. ZAR1 sequesters dormant mRNAs in immature oocytes, thereby preventing their translation. Importantly, ZAR1 undergoes proteasomal degradation during meiotic maturation of the oocyte, thereby leading to MARDO dissolution (Cheng et al., 2022). How ZAR1 is specifically degraded during oocyte maturation remained unknown.

In this manuscript, Fan and colleagues investigated the role of the E3 Ubiquitin ligase adaptor KLHL8 during female meiosis in mice. They found that KLHL8 deletion in oocytes leads to complete female infertility characterized by severe oocyte defects during meiotic maturation. At the molecular level, the authors show that KLHL8 leads to the excessive accumulation of the MARDO protein ZAR1 in oocytes, and to the consequent stabilization of several mRNAs during meiotic maturation. Through biochemical characterizations using overexpressed proteins in cultured cells, they further propose that KLHL8 acts as a substrate adaptor to mediate ZAR1 ubiquitination and proteasomal degradation.

Thus, this study adds a molecular mechanism involving KLHL8 to explain ZAR1 degradation and the dissolution of the MARDO during oocyte maturation. These findings are highly relevant, as they uncover a physiological mechanism for regulating the levels of ZAR1, an essential protein, and consequently of the maternal mRNAs in the oocyte.

Overall, the data appear sound and the conclusions mostly coherent with the data. The study would be significantly strengthened if the authors could repeat some of their experiments using additional controls and/or better experimental setups, as detailed below.

Major points:

- The title is misleading. As correctly pointed out by the authors themselves (line 68 and subsequent), KLHL8 acts as a substrate adaptor, not as E3 ubiquitin ligase itself. A better title would be "The E3 ubiquitin ligase adaptor KLHL8 targets ZAR1 to regulate maternal mRNA degradation in oocytes".

- Figure 4D: This experiment would be strongly strengthened if the authors could repeat it with the KLHL8 mutant that does not co-precipitate ZAR1, and show that ZAR1 stability is not affected.

- Figure 4F: The Ub-ZAR1 blot is too saturated to judge whether there is any effective difference in presence of KLHL8. Also, similar to the point above, it would be essential to repeat the experiment with the KLHL8 mutant that does not co-precipitate ZAR1, given that the assay is performed in cells and many indirect effects can occur. Indeed, strictly speaking this is not an in vitro ubiquitination assay, which would require the use of purified E1/E2/E3 ligases and/or substrates. I therefore also advise to change the definition "in vitro ubiquitination assay" in line 156. Finally, the antibodies used for IP and WB should be indicated in the interest of clarity. I.e., is the upper part of the blot labelled as "Ub-ZAR1" is a Ubiquitin blot or a ZAR1/FLAG blot? The same applies to all other IP/WB panels.

- Figure 5F: JC-1 has been shown to be highly prone to artefacts in mouse oocytes (PMID: 31579926). In particular, the subcortical high-membrane potential domain typical of JC-1 staining in oocytes (also visible in this picture) was not reproduced

by TMRE. I would strongly advise the authors to repeat the experiment using TMRE instead of JC-1. It would also be nice to see a panel with a zoom-in on one oocyte next to low-magnification images of several oocytes at once.

Minor points:

- In the interest of improving data interpretation by the reader, it would be advisable to rephrase some figure legends with more objective descriptions of what is represented. For instance, in Fig. 4F "KLHL8 promoted the ubiquitination-mediated protein degradation of ZAR1 in cultured HeLa cells" should be changed to, e.g., "FLAG IP of HeLa cells transfected with ZAR1-FLAG and with or without KLHL8-HA, followed by immunoblot for Ub".
- Line 150: Co-IPs from cell lysate don't allow to confirm a direct interaction between proteins. Even in presence of truncation mutants of either binding partner that abolish co-precipitation, it can never be formally excluded that interaction doesn't occur through a third unknown binding partner. If the authors want to establish whether KLHL8-ZAR1 interaction is direct, they should perform in vitro assays (GST-pulldown or similar) with both proteins recombinantly expressed and purified, to exclude that any third interaction partner mediates the interaction. Otherwise, I strongly advise to rephrase the sentence eliminating "and direct".
- Lines 117-118: "Although a small proportion of Klhl8^{oo-/-} oocytes could extrude the first polar body, these still failed to be fertilized (Fig. 2L-M)". How is the spindle morphology in these few oocytes that manage to extrude PB1 in vivo?
- Figure 3B: there are two instances of p-values equal to 0.0000. Can the authors indicate the actual values with numbers in scientific notation?
- Figure 3C: A calibration bar for the heatmap is missing.
- Lines 142-143: "Co-immunoprecipitation in HeLa cells showed that KLHL8 interacted with ZAR1 but not YBX2 (Fig. 4A)". The actual data don't seem to support this statement, as there is a clear band at the expected height in the YBX2 blot.
- Figure S6: Actually, the Golgi distribution seems to be affected by KLHL8 deletion, at least in this picture. What are the two cortical blobs visible in in the KO oocyte? Are they occurring in all oocytes? Are they co-localizing with mitochondria/MARDO? To facilitate interpretation of the data, I would also strongly advise the authors to include the specific proteins immunolabeled to mark each organelle either in the figure legend or next to the images themselves (as done in Fig. S7, for instance).
- Figure 6A-G: Does KLHL8 overexpression in WT oocytes lead to the excessive degradation of endogenous ZAR1 and MARDO premature dissolution? Similarly, how do WT oocytes injected with KLHL8 mRNA compare with mock injected WT oocytes in terms of PB1 extrusion rate?
- Figure 7E: The graph Y axis indicates "Relative intensity". What are the data normalized to?
- Line 229: "We found significant overlap between the proteins that accumulated in Klhl8^{oo-/-} GV oocytes and those that are degraded when transitioning from the GV to MII stage (Figure S8)". "Significant" implies that a statistical significance test has been performed, showing a p-value below a given threshold. It doesn't seem that a statistical test has been performed here, so I would refrain from use the word "significant".
- Line 241: "Together these findings suggest that the KLHL8-ZAR1-MARDO pathway is specific to oocytes". This conclusion is obvious, given that ZAR1 is an oocyte-specific protein (PMID: 12539046). Would KLHL8 co-localize with mitochondria in somatic cells overexpressing ZAR1?
- Fig. 5E and 5H-I can be moved to supplementary information.

Thanks for the suggestions and comments from three reviewers and Editor, which will greatly improve our manuscript. Based on suggestions, we performed series of additional experiments, including the protein level of KLHL8 in oocytes and embryos (Fig.1B), quantification of KLHL8 protein level (Fig.EV1B), a repeat of DDX6 protein level detection(Fig.3H), the spindle morphology in *Klhl8^{oo-/-}* MII oocytes (Fig.EV2), translation efficiency in *Klhl8^{oo-/-}* oocytes (Fig.EV3), the effect of KLHL8 mutant on ZAR1 stability and ubiquitination (Fig.EV4C-F), the protein level of ZFP36L2 ,ZAR2, LSM14B and G3BP2 in control and *Klhl8^{oo-/-}* oocytes (Fig. EV5C-J), the MPF activity at GV and Telophase I stage (Appendix Fig. S3), the mitochondrial membrane potential (Appendix Fig. S5A-B), the localization of KLHL8 in overexpressing ZAR1 HeLa cells (Appendix Fig. S7). Also, we carefully revised the statements and tone down the conclusions. The following are specific responses.

Responses to reviewers:

Referee #1:

Despite the recent progresses in clarifying maternal transcript homeostasis during oocyte maturation and maternal-to-zygotic transition in mammals, the equally important maternal protein homeostasis, which is essential for meiotic cell-cycle progression and subsequent embryonic development, remain to be investigated in depth. In this study, the authors identified a new maternally enriched E3 ubiquitin ligase, KLHL8, is highly expressed in mouse oocytes and co-localizes with mitochondria. The functional importance of this maternal protein was demonstrated by the fact that oocyte-specific deletion of *Klhl8* causes female infertility due to oocyte maturation defects. The results of mechanistic studies showed that ZAR1, an RNA binding protein which is required for mitochondria-associated ribonucleoprotein domain (MARDO) dissolution, is specifically recognized and degraded by KLHL8-mediated ubiquitination. In general, this manuscript is clearly written. The experiments

were well designed, and the results appropriately presented. I have some suggestions and comments to potentially improve the current manuscript.

Response: Thanks for the suggestions and comments, which will greatly improve our manuscript.

1. Author proved that KLHL8 target to ZAR1 and prevent the dissolution of MARDO, which caused the oocytes retardation in MI stage. However, according to the previous work (Cheng et.al, Science, 2022; Figure 4F), although the over-expression of mZar1 indeed delayed MARDO dissolution, the extrusion of PB1 was not significantly influenced. The results seem not consistent with the phenotype of *Klhl8*^{00-/-} oocytes in Figure 2H. As there are many other differentially expressed proteins were detected in KLHL8 deleted oocytes, there may be other key substrates that cause the developmental retardation and infertility. More discussion is needed about this.

Response: Thank you for the suggestion. According to the previous work, MARDO is assembled during oocyte growth (Science. 2022,378 (6617): eabq4835). Meanwhile, ZAR1 is essential for the maternal mRNA stability in growing oocytes (Cell Mol Life Sci. 2022,79 (2):92). These results indicate that MARDO aggregation and ZAR1 accumulation induced by KLHL8 deletion might be occurred during the growing stage of oogenesis. Therefore, overexpression of ZAR1 in the GV stage may be too late to affect the first polar body extrusion. In the following study, we will mimic the phenotype by constructing ZAR1-ZP3 overexpressing mice.

We also agreed with the opinion that there may be other key substrates that cause the developmental retardation and infertility and discussed other potential targets in the discussion part as following: “These differentially expressed proteins might be responsible for the developmental retardation and infertility. DAZL and MARF1 were also accumulated in *Klhl8* deleted oocytes (Fig. 3C and Fig.EV5A-B). DAZL, a key RNA-binding protein that has been reported to control maternal mRNA translation (Chen et al, 2011; Fukuda et al, 2018), is involved in the

development of female germ cells, and overexpression of DAZL leads to preimplantation developmental defects (Fukuda et al, 2018). MARF1 is essential for meiotic progression and female fertility. Mutations in *MARF1* cause female infertility characterized by up-regulation of transcripts, defective cytoplasmic maturation, and meiotic arrest (Su et al, 2012). While, the protein level of ZFP36L2 was obvious decreased in *Klh18^{oo/-}* oocytes (Fig. EV5C-D). ZFP36L2 is important for transcriptional silence in oocytes (Chousal et al, 2018), and *Zfp36l2* depletion results in oocyte maturation defects and aberrant spindle assembly (Sha et al, 2018).”

2. Authors proved that over-expressing *Klh18* can decrease the defects in *Klh18^{oo/-}* oocytes, and declared the importance of KLHL8-ZAR1-MARDO pathway. However, the results are not sufficient to support the importance of ZAR1 degradation to the meiosis. Deletion of ZAR1 in *Klh18^{oo/-}* oocytes is needed.

Response: Based on the suggestions, we tried to delete ZAR1 in *Klh18^{oo/-}* oocytes by Trim away (*Cell*. 2017, 171(7): 1692-1706). However, due to the effectiveness of antibody, we could not successfully knockdown the ZAR1 (Fig A). And the positive control TUBG could be knock down successfully (Fig B). We also attempted to downregulate ZAR1 by siRNA injection, but this method is still not work (Fig C). And, we will further explore this by producing a more effective antibody in the future.

ZAR1 Knockout in GV oocytes through Trim away and siRNA. (A) Representative immunofluorescence images of MARDO in WT and *Khl8^{oo-/-}* oocytes with or without ZAR1 antibody injection. (B) Representative immunofluorescence images of spindle in WT and *Khl8^{oo-/-}* oocytes with or without TUBG antibody injection. (C) Representative immunofluorescence images of MARDO in WT and *Khl8^{oo-/-}* oocytes with or without ZAR1-siRNA injection.

3. According to Gabriele et.al (Cell, 2024), there is the other organelle containing proteasomal system in oocytes named ELVA that can selectively degrade protein aggregates during meiosis. As the mitochondrion are excluded from this structure, and ZAR1, LSM14B and G3BP2 had been confirmed not to be degraded by ELVAs, it would be interesting to discuss the relationship KLHL8-dependent ub-proteasomal system and ELVAs in oocytes. And the abundance of other two proteins (LSM14B and G3BP2) can also be detected in *Khl8^{oo-/-}* oocytes.

Response: Thanks for your suggestions. We detected the protein level of

LSM14B and G3BP2 in *Klh18*^{oo-/-} oocytes, and their abundance did not change after *Klh18* deletion (Fig. EV5G-J). We added these results in the revised manuscript.

We also discussed the relationship KLHL8-dependent ub-proteasomal system and ELVAs as following: “Endolysosomal vesicular assemblies (ELVAs) are non-membrane-bound compartments composed of endolysosomes, autophagosomes, and proteasomes, which sequester and degrade aggregated proteins in oocytes (Zaffagnini et al, 2024). KLHL8 localized in the mitochondrial region and promoted proteins degradation through ubiquitination pathway, which is distinct from ELVA. The degradation activity of ELVAs is activated after oocyte maturation. However, KLHL8 is weakly expressed and localized in MII oocytes. We also detected the abundance of LSM14B and G3BP2 which had been confirmed not to be degraded by ELVAs. There was also no significant difference in their protein levels between WT and *Klh18*^{oo-/-} oocytes (Fig. EV5G-J).”

4. Is ZAR2, or ZAR1L, the genetic and functional homolog of ZAR1 in mouse oocytes but was not being paid sufficient attentions as ZAR1, also a target of KLHL8 for degradation? This protein was not mentioned in the manuscript. But in my opinion, at least some descriptions of ZAR2 should be added in the Introduction, and its possibility as a KLHL8 target is needed to be discussed.

Response: Thank you for the suggestion. We detected the protein level of ZAR2 in *Klh18*^{oo-/-} oocytes, and its abundance did not change after *Klh18* deletion. We have added the result in the revised manuscript as following : “We performed western blot assay and found that the protein abundance of ZAR2 was not affected after *Klh18* deletion (Fig. EV5E-F).”

We also introduced ZAR2 as following: “For example, many RNA binding proteins are significantly decreased during meiosis, like ZAR1, ZAR2, YBX2, and PATL2, which are responsible for maternal mRNA stability and

metabolism (Rong et al, 2019; Wu & Fan, 2022; Zhang et al, 2023c, 2023b).” and discussed its possibility as a KLHL8 target: “*Zar1* and *Zar2* was reported partial functional redundancy in multiple studies (Wu & Fan, 2022; Rong et al, 2025). We performed western blot assay and found that the protein abundance of ZAR2 was not affected after *Klhl8* deletion (Fig. EV5E-F). It indicates that ZAR2 is unlikely to be a target of KLHL8.”

5. In Figure 1B, the protein abundance of KLHL8 is specifically low in GVBD stage oocytes compared to GV or MI stage oocytes. Are there any physiological or technological explanations for this phenomenon?

Response: Thanks for pointing out this point. The band of KLHL8 in GVBD oocyte is weaker than GV and MI stage oocytes was due to the variation of internal control. We repeated the assay (Fig. 1B) and provided quantitative results (Fig. EV1B) in the revised manuscript. The protein abundance of KLHL8 increases after GVBD and reach the peak at MI stage.

6. In Figure 2E and line 112, there are many abnormal cytoplasmic granules in *Klhl8*^{00/-} oocytes, what are the component of these granules?

Response: According previous reports, the dark and granular cytoplasm was usually caused by abnormal organelle distribution (*J Biochem.* 2020, 167(3): 257–266; *Curr. Biol.* 2014, 24(20): 2451–2458). According to our immunofluorescence and transmission electron microscopy data, it is mitochondria and MARDO aggregates (Fig. 5A-C).

7. In line 133-134, it was confirmed that the protein of ZAR1 is sharply increased in *Klhl8*^{00/-} oocytes with mRNA unchanged. Except the explanation that KLHL8 help with degrading ZAR1, the translation efficiency is also needed to be detected.

Response: We detected the translation efficiency by GFP mRNA injection in WT and *Klhl8*^{00/-} oocytes. There is no difference in translation efficiency between control and *Klhl8* deletion oocytes (Fig. EV3).

We added the result in the revised manuscript as following: “There is no difference in translation efficiency between WT and *Klh18*^{00/-} oocytes (Fig. EV3), indicating that the accumulation of ZAR1 was due to the protein degradation.”

8. In line 173 and Figure 5C, what is 'mitochondrial cluster index'? Please explain.

Response: 'Mitochondrial cluster index' is an indicator for evaluating mitochondrial aggregation from the previous work (*Science*. 2022,378 (6617): eabq4835), which was assessed by calculating the reciprocal of the mitochondrial cluster number.

9. In line 203-204, the relative RNA copy numbers normalized by spike-in or reference genes is needed to support the abnormal RNA abundance.

Response: Thank you for the suggestion. EGFP is spike-in gene of RNA sequencing, and its levels of the two groups are similar. We added new sheets normalized by spike-in gene in Dataset EV2.

10. In Figure 4A, 'ZAR1-FALG' should be 'ZAR1-FLAG'.

Response: We revised 'ZAR1-FALG' to 'ZAR1-FLAG' in Figure 4A.

11. In Figure 4C, the marker and the number of ZAR1-FLAG_input did not correspond, the same in Figure 4F_ZAR1-FLAG_IP, the band of ZAR1-FLAG_IP is too low. Please check.

Response: We revised this in the revised manuscript.

12. In the figure legend of Figure 4F, authors declared that KLHL8 promoted the ubiquitination-mediated protein degradation of ZAR1. However, this figure just proved the KLHL8-dependent ubiquitination of ZAR1 but not the degradation. It needs to be more precise.

Response: Thanks for your suggestion. We revised the conclusion to “KLHL8 promoted the ubiquitination of ZAR1 in HeLa cells.”

13. In Figure 5F, the JC-1-stained mitochondrion seemed to be circles, which is unusual. Please explain the mitochondrial morphology.

Response: This is the characteristic of JC-1 with a bright cortical ring of red J-aggregate fluorescence (*Mol Hum Reprod.* 2019, 25(11):695-705), so the signal seems to be circles. Based on the suggestion from another reviewer, we repeat the experiment using TMRE instead of JC-1 (Appendix Fig. S5A-B).

We updated the result in the revised manuscript as following: “However, oocyte deletion of *Klh18* had no effect on mitochondrial membrane potential (MMP), reactive oxygen species (ROS) or ATP levels (Appendix Fig. S5A-E).”

14. Some pertinent references should be cited during revision. For example, a review article comparing the homology and redundancy of ZAR1 and ZAR2 (PMID: 35072788), and a newly published study showing the association of ZAR1/2-mediated epigenetic regulations with oocyte aging (PMID: 39755931).

Response: Thank you for the suggestion. We cited these references in revised manuscript.

Referee #2:

The authors present an interesting manuscript on the role of the KLHL8 involved in the ubiquitination of proteins in the regulation of oocyte meiotic progression. They have shown that the absence of KLHL8 leads to female sterility due to the lack of extrusion of the polar body. The study is very interesting and contributes to the molecular physiology of oocytes. However, it needs to be significantly improved before publication.

Response: Thank you for your interest and recognition to the study, and thank you for providing suggestions to improve the study. We added additional experiments and evidences in our manuscript and toned-down some of the conclusions. Below are our responses to all comments point by point.

1. Fig.1B Please provide quantitatively equal samples to describe expression of KLHL in oocytes. Based on Vinculin authors load different amount of cell in the stages.

Response: Thanks for indicating this. We repeated the assay (Fig. 1B) and provided quantitative results (Fig. EV1B) in the revised manuscript.

2. Where is KLHL8 localized?

Response: According to Fig. 1C, immunofluorescent staining revealed co-localization of KLHL8 and mitochondria in mouse oocyte.

3. Fig.2EG Presentation of data not convincing. Provide, clear BF images of better quality with equal light intensity and contrast.

Response: We provided clear BF images of better quality in Fig.2EL

4. Fig.2GH Image and quantification doesn't positively correlates in term of PBE.

Response: We provided the images of GVBD and PB1 in Fig.2G corresponding to the quantification results.

5. Does cytoplasmic dysmorphism refer to cytoplasm granulation? This morphology is common for C57BL/6J genetic background.

Response: Cytoplasmic dysmorphism usually refer to appearance of translucent vacuoles and dark, spongy granulated area in the cytoplasm. The appearance of dark and granulated area in the cytoplasm is considered as organelle clustering (*Hum Reprod.* 2008,23(8):1778-85; *J Biochem.* 2020, 167(3): 257–266). Compared to ICR mice, C57BL mice oocytes exhibit physiological cytoplasmic granulation. However, *Klh18* deletion leads to organelles aggregation, which is significantly different from physiological cytoplasmic granulation. According to our immunofluorescence and transmission electron microscopy data, the mitochondria are severe aggregated (Fig. 5A-C).

6. Additional quantification of PNs appears as essential based on statement in

Results section and Fig.2L.

Response: We provided the 2PNs quantification in Fig.2M.

7. The absence of MII oocyte indicate MI arrest in which MPF plays a key role. Does permanent activity of MPF in the *KLHL8*^{-/-} oocytes leads to reported phenotype?

Response: Compared with WT oocytes, the activity of MPF (Cyclin B1 and p-CDK1) was not affected in *Klh18*^{oo-/-} oocytes. Thus, it is not permanent activity of MPF that leads to maturation defects in *Klh18*^{oo-/-} oocytes. We added the result in the revised manuscript as following : “Immunoblotting demonstrated that the activity of MPF was not affected after *Klh18* deletion (Appendix Fig. S3A-B).”

8. Fig.5F fluorescence signal is not visible.

Response: Thank you for the suggestion. Based on the suggestion from another reviewer, we repeated the experiment using TMRE instead of JC-1 (Appendix Fig. S5A-B). We updated the result in the revised manuscript as following: “However, oocyte deletion of *Klh18* had no effect on mitochondrial membrane potential (MMP), reactive oxygen species (ROS) or ATP levels (Appendix Fig. S5A-E).”

9. As expected, the authors conclude that the stability of ZAR1 is regulated by UPS. However, the conclusion that KLHL8 is responsible for the degradation of ZAR1 was not confirmed in the oocyte.

Response: Thank you for the question. Co-transfection experiments have shown that KHL8 promotes degradation of ZAR1 in HeLa cells (Fig. 4D, E). The ubiquitination experiment demonstrated that KHL8 promotes the ubiquitination of ZAR1 (Fig. 4F). Immunoblotting result showed that *Klh18* deletion led to ZAR1 accumulation (Fig. 3D, E). And *Klh18* mRNA rescue experiments demonstrated that KLHL8 promotes ZAR1 protein degradation *in vivo* (Fig. 6A, B). We also overexpressed KLHL8 mutant in

Klh18^{00/-} oocytes, but the mutant was failure to promote ZAR1 degradation (Fig. EV4F). Furthermore, *Klh18* deletion does not affect mRNA transcription (Fig. 3F) and translation (Fig. EV3). These results collectively suggest that KLHL8 promotes the degradation of ZAR1 through ubiquitination. Therefore, it can be explained that KLHL8 is responsible for the degradation of ZAR1 in oocytes.

10. Authors state that morphology of ER or Golgi is equal between genotypes (Fig.S6). Morphology/staining of ER or Golgi is unusual based what has been reported in literature.

Response: ER and Golgi apparatus of oocytes are different from those of somatic cells and changed with different stages. ER and Golgi apparatus exhibit reticular and punctate morphology in mouse GV stage oocytes as reported in several studies (*Dev Biol.* 2007, 305(1):133-44; *Reprod Biol Endocrinol.* 2013, 19:11:31; *Science.* 2022, 378(6617): eabq4835). Due to insufficient specificity of GM130 antibody, abnormal signals of mitochondrial aggregation appeared in *Klh18*^{00/-} oocytes. And this part had less relevant to the main conclusions, we deleted it in the revised manuscript.

11. In the Volcano plots Fig.7 highlight/point transcripts analyzed in this study. L207 Authors state that KLHL regulate transcriptome prior GV, please explain the conclusion by which data is supported.

Response: Thank you for the suggestion. Due to the lack of direct evidence, we deleted this sentence "These results indicated that KLHL8 regulated the maternal transcriptome prior to the GV stage".

12. Additionally, due to absence of MII oocytes in the *Klh18*^{00/-} mice (Fig.2G) which groups were compared for DE (differential expression) here?

Response: About 15% MII oocytes could be obtained from *Klh18*^{00/-} mice. These MII oocytes were used for RNA sequencing.

13. Does RNA-seq and proteomic datasets correlate?

Response: There is a correlation between RNA-seq and proteomic datasets. *Klhl8* deletion caused accumulation of RNA binding proteins in proteomic datasets. The accumulation of RNA binding proteins like ZAR1 could cause mRNA accumulation.

12. L256 Statement is not supported by the provided data and appears as speculative.

Response: Thanks for your suggestion, we revised this paragraph as following: “Collectively, this study suggests KLHL8 was essential for female fertility by regulating the maternal protein homeostasis in mouse oocytes.”

13. Discuss what has been presented in relation to ubiquitination and RNA metabolism.

Response: In this study, the relationship between ubiquitination and mRNA metabolism is linked through RNA binding proteins. ZAR1 regulates mRNA metabolism by modulating MARDO dissolution, while KLHL8 can regulate ZAR1 degradation through ubiquitination. Thus, KLHL8 indirectly affects mRNA metabolism by promoting the ubiquitination degradation of maternal RNA binding protein.

We added the discussion in the revised manuscript as following: “The relationship between ubiquitination and mRNA metabolism is linked through RNA binding proteins. ZAR1 regulates mRNA metabolism by modulating MARDO dissolution, while KLHL8 can regulate ZAR1 degradation through ubiquitination. Thus, KLHL8 indirectly affects mRNA metabolism by promoting the ubiquitination degradation of maternal RNA binding protein. This study thus suggests a novel role for the ubiquitin-proteasome pathway in both maternal protein and mRNA clearance during oocyte maturation.”

14. What is the first band in the IB images?

Response: There is a protein ladder. To better understanding for readers, we use “M” to label this protein ladder in revised manuscript.

15. Absence of number of cells analyzed in experiments.

Response: Thanks for your suggestion, we indicated the number of cells in the figure legends of relevant data.

Referee #3:

The regulation of mRNA stability during oocyte maturation and early embryo development is essential to ensure proper embryonic development. In 2022, Cheng et al. identified a mitochondria-associated RNA domain (MARDO) in mammalian oocytes which is assembled by the oocyte-specific protein ZAR1. ZAR1 sequesters dormant mRNAs in immature oocytes, thereby preventing their translation. Importantly, ZAR1 undergoes proteasomal degradation during meiotic maturation of the oocyte, thereby leading to MARDO dissolution (Cheng et al., 2022). How ZAR1 is specifically degraded during oocyte maturation remained unknown. In this manuscript, Fan and colleagues investigated the role of the E3 Ubiquitin ligase adaptor KLHL8 during female meiosis in mice. They found that KLHL8 deletion in oocytes leads to complete female infertility characterized by severe oocyte defects during meiotic maturation. At the molecular level, the authors show that KLHL8 leads to the excessive accumulation of the MARDO protein ZAR1 in oocytes, and to the consequent stabilization of several mRNAs during meiotic maturation. Through biochemical characterizations using overexpressed proteins in cultured cells, they further propose that KLHL8 acts as a substrate adaptor to mediate ZAR1 ubiquitination and proteasomal degradation. Thus, this study adds a molecular mechanism involving KLHL8 to explain ZAR1 degradation and the dissolution of the MARDO during oocyte maturation. These findings are highly relevant, as they uncover a physiological mechanism for regulating the levels of ZAR1, an

essential protein, and consequently of the maternal mRNAs in the oocyte.

Overall, the data appear sound and the conclusions mostly coherent with the data. The study would be significantly strengthened if the authors could repeat some of their experiments using additional controls and/or better experimental setups, as detailed below.

Response: Thanks for your recognition in our work. And we believe your suggestions will improve the quality of the manuscript. Following were our responses point by point.

1. - The title is misleading. As correctly pointed out by the authors themselves (line 68 and subsequent), KLHL8 acts as a substrate adaptor, not as E3 ubiquitin ligase itself. A better title would be "The E3 ubiquitin ligase adaptor KLHL8 targets ZAR1 to regulate maternal mRNA degradation in oocytes".

Response: We agreed with your opinion. We changed the title as "The E3 ubiquitin ligase adaptor KLHL8 targets ZAR1 to regulate maternal mRNA degradation in oocytes".

2. - Figure 4D: This experiment would be strongly strengthened if the authors could repeat it with the KLHL8 mutant that does not co-precipitate ZAR1, and show that ZAR1 stability is not affected.

Figure 4F: The Ub-ZAR1 blot is too saturated to judge whether there is any effective difference in presence of KLHL8. Also, similar to the point above, it would be essential to repeat the experiment with the KLHL8 mutant that does not co-precipitate ZAR1, given that the assay is performed in cells and many indirect effects can occur.

Response: We evaluated the KLHL8 mutant on ZAR1 stability and ZAR1 ubiquitination. Compared with KLHL8 WT, KLHL8 mutant overexpression does not affect ZAR1 stability both in HeLa cells and oocytes (Fig. EV4C-D, F). KLHL8 mutant does not affect the ubiquitination level of ZAR1 (Fig. EV4E).

We added the result in the revised manuscript as following: "KLHL8

mutants with BACK domain deletion lost the ability to promote the degradation and ubiquitination of ZAR1 (Fig. EV4C-E).” and “As predicted, the protein level of ZAR1 decreased in *Klh18^{oo-/-}* GV oocytes after exogenous *Klh18* mRNA supplementation compared to noninjected and *Klh18-ΔBACK* mRNA injection oocytes (Fig. 6A, B; Fig. EV4F).”

3. - Indeed, strictly speaking this is not an in vitro ubiquitination assay, which would require the use of purified E1/E2/E3 ligases and/or substrates. I therefore also advise to change the definition "in vitro ubiquitination assay" in line 156. Finally, the antibodies used for IP and WB should be indicated in the interest of clarity. I.e., is the upper part of the blot labelled as "Ub-ZAR1" is a Ubiquitin blot or a ZAR1/FLAG blot? The same applies to all other IP/WB panels.

Response: We changed the definition "*in vitro* ubiquitination assay" to "*in vivo* ubiquitination assay". The upper part of the blot is a Ubiquitin blot, and we changed the label as "Ubiquitin". We also checked all the figures and adjusted the descriptions of other IP/WB panels.

4. - Figure 5F: JC-1 has been shown to be highly prone to artefacts in mouse oocytes (PMID: 31579926). In particular, the subcortical high-membrane potential domain typical of JC-1 staining in oocytes (also visible in this picture) was not reproduced by TMRE. I would strongly advise the authors to repeat the experiment using TMRE instead of JC-1. It would also be nice to see a panel with a zoom-in on one oocyte next to low-magnification images of several oocytes at once.

Response: Thank you for the suggestion. We repeated the experiment using TMRE instead of JC-1. There was no significant difference in fluorescence intensity between WT and *Klh18^{oo-/-}* oocytes (Appendix Fig. S5A-B). We updated the result in the revised manuscript as following: “However, oocyte deletion of *Klh18* had no effect on mitochondrial membrane potential (MMP), reactive oxygen species (ROS) or ATP levels (Appendix Fig. S5A-E).”

4. - In the interest of improving data interpretation by the reader, it would be advisable to rephrase some figure legends with more objective descriptions of what is represented. For instance, in Fig. 4F "KLHL8 promoted the ubiquitination-mediated protein degradation of ZAR1 in cultured HeLa cells" should be changed to, e.g., "FLAG IP of HeLa cells transfected with ZAR1-FLAG and with or without KLHL8-HA, followed by immunoblot for Ub".

Response: Based on the suggestion from the reviewer, we changed Fig.4F legend as "FLAG IP of HeLa cells transfected with ZAR1-FLAG and with or without KLHL8-HA, followed by immunoblot for ubiquitin" and checked the descriptions in other figure legends.

5. - Line 150: Co-IPs from cell lysate don't allow to confirm a direct interaction between proteins. Even in presence of truncation mutants of either binding partner that abolish co-precipitation, it can never be formally excluded that interaction doesn't occur through a third unknown binding partner. If the authors want to establish whether KLHL8-ZAR1 interaction is direct, they should perform in vitro assays (GST-pulldown or similar) with both proteins recombinantly expressed and purified, to exclude that any third interaction partner mediates the interaction. Otherwise, I strongly advise to rephrase the sentence eliminating "and direct".

Response: Thank you for indicating this, we deleted "and direct" in revised manuscript.

6. - Lines 117-118: "Although a small proportion of *Klh18*^{00/-} oocytes could

extrude the first polar body, these still failed to be fertilized (Fig. 2L-M)". How is the spindle morphology in these few oocytes that manage to extrude PB1 in vivo?

Response: We checked the spindle morphology of few retrieved MII oocytes. Most of them showed absent spindles, which entered the first polar body without division and exhibit abnormal morphology. A small portion of oocytes had obvious cytoplasmic fragmentation and abnormal spindle morphology that are not properly separated (Fig. EV2A, B).

We added the result in the revised manuscript as following: "Most of them showed absent spindles, which entered the first polar body without division and exhibit abnormal morphology. A small portion of oocytes had obvious cytoplasmic fragmentation and abnormal spindle morphology that are not properly separated (Fig. EV2A, B)."

7. - Figure 3B: there are two instances of p-values equal to 0.0000. Can the authors indicate the actual values with numbers in scientific notation?

Response: There are two p-values equal to 0.00001 and 0.000001. We provided the actual p-values.

8. - Figure 3C: A calibration bar for the heatmap is missing.

Response: We added the calibration bar for the heatmap in the revised manuscript.

9. - Lines 142-143: "Co-immunoprecipitation in HeLa cells showed that KLHL8 interacted with ZAR1 but not YBX2 (Fig. 4A)". The actual data don't seem to support this statement, as there is a clear band at the expected height in the YBX2 blot.

Response: There was a weak interaction between KLHL8 and YBX2, and we updated the conclusion as following: " Co-immunoprecipitation in HeLa cells showed that KLHL8 interacted with ZAR1 clearly but had a very weak interaction with YBX2 (Fig. 4A)".

10. - Figure S6: Actually, the Golgi distribution seems to be affected by KLHL8 deletion, at least in this picture. What are the two cortical blobs visible in the KO oocyte? Are they occurring in all oocytes? Are they co-localizing with mitochondria/MARDO? To facilitate interpretation of the data, I would also strongly advise the authors to include the specific proteins immunolabeled to mark each organelle either in the figure legend or next to the images themselves (as done in Fig. S7, for instance).

Response: Thank you for the suggestion. The cortical spots in *Klhl8*^{oo-/-} oocytes are signals of mitochondrial aggregation caused by insufficient specificity of GM130 antibody, and they exist in all *Klhl8*^{oo-/-} oocytes. And this part had less relevant to the main conclusions, we deleted it in the revised manuscript.

11. - Figure 6A-G: Does KLHL8 overexpression in WT oocytes lead to the excessive degradation of endogenous ZAR1 and MARDO premature dissolution? Similarly, how do WT oocytes injected with KLHL8 mRNA compare with mock injected WT oocytes in terms of PB1 extrusion rate?

Response: Actually, we supplemented KLHL8 mRNA in WT oocytes, but it did not affect PB1 extrusion rate compared with WT oocytes (Fig.6F, G). And it didn't cause excessive degradation of endogenous ZAR1 and MARDO premature dissolution in WT oocytes (Fig.6A-E). This indicated endogenous KLHL8 was sufficient for its function.

12. - Figure 7E: The graph Y axis indicates "Relative intensity". What are the data normalized to?

Response: We directly measured fluorescence intensity and subtracted the background signal. Therefore, we changed " Relative intensity " to " Intensity " in the graph Y axis.

13. - Line 229: "We found significant overlap between the proteins that accumulated in *Klhl8*^{oo-/-} GV oocytes and those that are degraded when transitioning from the GV to MII stage (Figure S8)". "Significant" implies that a

statistical significance test has been performed, showing a p-value below a given threshold. It doesn't seem that a statistical test has been performed here, so I would refrain from use the word "significant".

Response: Thank you for the suggestion. I changed the word "significant" to "obvious".

14. - Line 241: "Together these findings suggest that the KLHL8-ZAR1-MARDO pathway is specific to oocytes". This conclusion is obvious, given that ZAR1 is an oocyte-specific protein (PMID: 12539046). Would KLHL8 co-localize with mitochondria in somatic cells overexpressing ZAR1?

Response: We checked the localization of KLHL8 and ZAR1 in HeLa cells. KLHL8 have a vesicular location, which is different from mitochondria and ZAR1 in HeLa cells. They do not co-localize with each other (Appendix Fig. S7A-B).

15. - Fig. 5E and 5H-I can be moved to supplementary information.

Response: Thank you for the suggestion. We moved Fig. 5E and 5H-I to supplementary information (Appendix Fig. S5C-E) in the revised manuscript.

Responses to the EMBO Reports Editorial Office

1. As you will see, the referees acknowledge that the findings are interesting and that the conclusions are overall supported by the data presented but they also raise a number of concerns and have suggestions how to further strengthen the data. I agree with all the points and concerns raised by the referees and these need to be addressed in full. It will be important to strengthen the data on ZAR1 as being a functionally relevant target downstream of KLHL8 and to at least discuss that other potential targets might be involved in the oocyte and infertility phenotype as well.

Response: Thank you for the suggestion. We discussed other potential targets in the discussion part as following: “These differentially expressed proteins might be responsible for the developmental retardation and infertility. DAZL and MARF1 were also accumulated in *Klh18* deleted oocytes (Fig. 3C; Fig. EV5A-B). DAZL, a key RNA-binding protein that has been reported to control maternal mRNA translation (Chen et al, 2011; Fukuda et al, 2018), is involved in the development of female germ cells, and overexpression of DAZL leads to preimplantation developmental defects (Fukuda et al, 2018). MARF1 is essential for meiotic progression and female fertility. Mutations in *MARF1* cause female infertility characterized by up-regulation of transcripts, defective cytoplasmic maturation, and meiotic arrest (Su et al, 2012). While, the protein level of ZFP36L2 was obvious decreased in *Klh18^{oo-/-}* oocytes (Fig. EV5C-D). ZFP36L2 is important for transcriptional silence in oocytes (Chousal et al, 2018), and *Zfp36l2* depletion results in oocyte maturation defects and aberrant spindle assembly (Sha et al, 2018).”

2. It is also important to test whether KLHL8 really degrades not only ubiquitinates ZAR1, also in the oocyte.

Response: We toned down the conclusions. Co-transfection experiments have shown that KHL8 promotes degradation of ZAR1 in HeLa cells (Fig. 4D, E). The ubiquitination experiment demonstrated that KHL8 promotes the ubiquitination of ZAR1 (Fig. 4F). Immunoblotting result showed that *Klh18* deletion led to ZAR1 accumulation (Fig. 3D, E). And *Klh18* mRNA rescue experiments demonstrated that KLHL8 promotes ZAR1 protein degradation *in vivo* (Fig. 6A, B). We also overexpressed KLHL8 mutant in *Klh18^{oo-/-}* oocytes, but the mutant was failure to promote ZAR1 degradation (Fig. EV4F). Simultaneously, *Klh18* deletion does not affect mRNA transcription (Fig. 3F) and translation (Fig. EV3). These results collectively suggest that KLHL8 promotes the degradation of ZAR1 through ubiquitination. However, there is no *in vitro* evidence that KLHL8 directly degrades ZAR1. Therefore, we have modified the statement

“KLHL8 promotes ZAR1 degradation” in the article.

IMPORTANT NOTE:

We perform an initial quality control of all revised manuscripts before re-review. Your manuscript will FAIL this control and the handling will be delayed IN CASE the following APPLIES:

1) A data availability section providing access to data deposited in public databases is missing. If you have not deposited any data, please add a sentence to the data availability section that explains that.

Response: Thank you for the suggestion. We provided Data availability section: “All the data have been included in the manuscript and supplementary data. RNA-seq raw sequence data reported in this paper have been deposited in the Genome Sequence Archive in National Genomics Data Center, China National Center for Bioinformation/Beijing Institute of Genomics, Chinese Academy of Sciences (GSA: CRA021060) that are publicly accessible at <https://ngdc.cncb.ac.cn/gsa>”.

2) Your manuscript contains statistics and error bars based on n=2. Please use scatter blots in these cases. No statistics should be calculated if n=2.

Response: We added a repeat in DDX6 protein level and updated the statistical chart in Fig.3H (n = 3).

=====

Response: We provided a .docx formatted version of the manuscript text and the changes are highlighted to be clearly visible.

2) individual production quality figure files as .eps, .tif, .jpg (one file per figure). Please download our Figure Preparation Guidelines (figure preparation pdf)

from our Author Guidelines pages <https://www.embopress.org/page/journal/14693178/authorguide> for more info on how to prepare your figures.

Response: We prepared figures as Figure Preparation Guidelines suggested.

Response: We provided a response letter including the reviewers' reports and detailed point-by-point responses to their comments.

4) a complete author checklist, which you can download from our author guidelines(<https://www.embopress.org/page/journal/14693178/authorguide>;). Please insert information in the checklist that is also reflected in the manuscript. The completed author checklist will also be part of the RPF.

Response: We provided completed author checklist.

5) Please note that all corresponding authors are required to supply an ORCID ID for their name upon submission of a revised manuscript (<https://orcid.org/>;). Please find instructions on how to link your ORCID ID to your account in our manuscript tracking system in our Author guidelines (<https://www.embopress.org/page/journal/14693178/authorguide#authorshipguidelines>);

Response: We supplied ORCID ID for all corresponding authors upon submission of a revised manuscript.

6) We replaced Supplementary Information with Expanded View (EV) Figures and Tables that are collapsible/expandable online. A maximum of 5 EV Figures can be typeset. EV Figures should be cited as 'Figure EV1, Figure EV2' etc... in the text and their respective legends should be included in the main text

after the legends of regular figures.

Response: Thank you for the suggestion. We added 5 EV Figures, and their legends were included in the main text after the legends of regular figures.

- For the figures that you do NOT wish to display as Expanded View figures, they should be bundled together with their legends in a single PDF file called *Appendix*, which should start with a short Table of Content. Appendix figures should be referred to in the main text as: "Appendix Figure S1, Appendix Figure S2" etc. See detailed instructions regarding expanded view here: <<https://www.embopress.org/page/journal/14693178/authorguide#expandedview>>;

Response: Thank you for the suggestion. We provided a short Table of Content and 7 Appendix figures in a single PDF file called *Appendix*.

Response: Thank you for the suggestion. Our additional Tables were labeled and referred to as Dataset EV1, EV2, etc. Legends were provided in a separate tab in case of .xls files.

7) Before submitting your revision, primary datasets (and computer code, where appropriate) produced in this study need to be deposited in an appropriate public database (see <<https://www.embopress.org/page/journal/14693178/authorguide#dataavailability>>). Specifically, we would kindly ask you to provide public access to the RNA-seq and proteomics datasets. Please remember to provide a reviewer password if the datasets are not yet public. The accession numbers and database should be listed in a formal "Data Availability " section (placed after Materials & Method) that follows the model below (see also

< <https://www.embopress.org/page/journal/14693178/authorguide#dataavailability>>);). Please note that the Data Availability Section is restricted to new primary data that are part of this study.

Data availability

- RNA-Seq data: Gene Expression Omnibus GSE46843
(<https://www.ncbi.nlm.nih.gov/geo/query/acc.cgi?acc=GSE46843>)

- [data type]: [name of the resource] [accession number/identifier/doi] ([URL or identifiers.org/DATABASE:ACCESSION])

*** Note - All links should resolve to a page where the data can be accessed.

Response: As you suggested, we provided public access to the RNA-seq and proteomics datasets. RNA-seq raw sequence data reported in this paper have been deposited in the Genome Sequence Archive in National Genomics Data Center, China National Center for Bioinformation/Beijing Institute of Genomics, Chinese Academy of Sciences (GSA: CRA021060) that are publicly accessible at <https://ngdc.cncb.ac.cn/gsa>.

8) At EMBO Press we ask authors to provide source data for the main figures. Our source data coordinator will contact you to discuss which figure panels we would need source data for and will also provide you with helpful tips on how to upload and organize the files. Additional information on source data and instruction on how to label the files are available <<https://www.embopress.org/page/journal/14693178/authorguide#sourcedata>>.

Response: Thank you for the suggestion. We provided source data for the main figures in submitted document.

9) The journal requires a statement specifying whether or not authors have competing interests (defined as all potential or actual interests that could be

perceived to influence the presentation or interpretation of an article). In case of competing interests, this must be specified in your disclosure statement.

Further information: <https://www.embopress.org/competing-interests>

Response: We provided a competing interests statement.

10) Figure legends and data quantification:

- the name of the statistical test used to generate error bars and P values,
- the number (n) of independent experiments (please specify technical or biological replicates) underlying each data point,
- the nature of the bars and error bars (s.d., s.e.m.)
- If the data are obtained from n {less than or equal to} 5, show the individual data points in addition to the SD or SEM.
- If the data are obtained from n {less than or equal to} 2, use scatter blots showing the individual data points.

Response: We checked and revised Figure legends and data quantification as you suggested.

Discussion of statistical methodology can be reported in the materials and methods section, but figure legends should contain a basic description of n, P and the test applied. See also the guidelines for figure legend preparation: <https://www.embopress.org/page/journal/14693178/authorguide#figureformat>

Response: We checked and revised Figure legends and scale bars of microscopy images as you suggested.

11) Our journal encourages inclusion of *data citations in the reference list* to directly cite datasets that were re-used and obtained from public databases. Data citations in the article text are distinct from normal bibliographical citations and should directly link to the database records from which the data can be accessed. In the main text, data citations are formatted as follows:

"Data ref: Smith et al, 2001" or "Data ref: NCBI Sequence Read Archive PRJNA342805, 2017". In the Reference list, data citations must be labeled with "[DATASET]". A data reference must provide the database name, accession number/identifiers and a resolvable link to the landing page from which the data can be accessed at the end of the reference. Further instructions are available at <https://www.embopress.org/page/journal/14693178/authorguide#referencesformat>;

Response: We provided data citation in the main text and Reference list as you suggested.

12) All Materials and Methods need to be described in the main text using our 'Structured Methods' format. According to this format, the Methods section includes a Reagents and Tools Table (listing key reagents, experimental models, software and relevant equipment and including their sources and relevant identifiers) followed by a Methods and Protocols section describing the methods, ideally using a step-by-step protocol format. The aim is to facilitate adoption of the methodologies across labs. Please download and fill our Reagents and Tools Table template (.docx), which you can find in our author

guidelines: <https://www.embopress.org/page/journal/14693178/authorguide#structuredmethods>. When submitting your revised manuscript, please do not include the Reagents and Tools Table in the Methods section of the manuscript but upload it as a separate file choosing the file type "Reagent Table".

An example of a Method paper with Structured Methods can be found here: <https://www.embopress.org/doi/10.15252/msb.20178071>.

Response: We have rewritten Materials and Methods part described in the main text by using 'Structured Methods' format.

Dear Dr. Zhang

Thank you for the submission of your revised manuscript to EMBO reports. We have now received the full set of referee reports that is copied below.

As you will see, all referees are very positive about the study and support publication. Before I can accept the manuscript, I need you to address some minor points below:

- Please remove the number of keywords to 5.
- Please update the 'Conflict of interest' paragraph to our new 'Disclosure and competing interests statement'. For more information see <https://www.embopress.org/page/journal/14693178/authorguide#conflictsofinterest>
- Please update the personal e-mail address of Dr. Zhihua Zhang to an institutional e-mail address in our online manuscript tracking system, since all communication with the corresponding author should be via the institutional address.
- Regarding the Author Contributions, we now use CRediT to specify the contributions of each author in the journal submission system. Therefore, please remove the Author Contributions from the manuscript file and make sure that the author contributions in our online manuscript tracking system are correct and up-to-date. The information you specified in the system will be automatically retrieved and typeset into the article. You can enter additional information in the free text box provided, if you wish.
- The Data Availability section should exclusively refer to data deposited in public repositories. Therefore, please remove the sentence "All the data have been included in the manuscript and supplementary data."
- Author Checklist, line 54: You state that information on cell line authentication or mycoplasma contamination testing is given in Materials and Methods, but I could not locate relevant statements. Please check.
- Please move the Ethics Approval statement to the Methods section.
- Materials and Methods should be Methods
- The manuscript sections should be in the following order: Title page - Abstract & Keywords - Introduction - Results - Discussion - Methods - Data Availability - Acknowledgments - Disclosure Statement & Competing Interests - References - Figure Legends - (Main Tables with legends if applicable) - Expanded View Figure Legends.
- Please provide the main and EV figures as separate production quality Figure files.
- Funding information: the funders in the Comments box in the online manuscript tracking system need to be removed and provided as separate entries: the Fund of Fudan University and Cao'ejiang Basic Research, the New Cornerstone Science Foundation through the XPLOER PRIZE, the Capacity Building Planning Program for Shanghai Women and Children's Health Service, and the Collaborative Innovation Center Project Construction for Shanghai Women and Children's Health.
- Can you please split the source data? We would need an individual zip file for each figure. Thank you.
- Source Data for Figure 1B "copy" could be called "replicate" and should be part of the folder for Figure 1B.
- Doing a spot check, I noticed that the Source Data .xls file for Figure 5D and 6E contain chinese characters for statistics points (correct?). I suggest replacing these with english characters to enhance readability.
- Could you please provide the Source Data for Appendix Figure S3B? Our image integrity software indicated a potential aberration, that we would like to clarify to avoid any ambiguities.
- Data reference to the dataset from Sun et al, 2023. The format is perfect, but you should also cite the study itself, i.e., (Data ref: Sun et al, 2023; Sun et al, 2023). The first citation is to the dataset (as you have it already in the reference list) and the second citation is to the published paper that reported the dataset.
- Our production/data editors have asked you to clarify several points in the figure legends (see below). Please incorporate these changes in the manuscript and return the revised file with tracked changes with your final manuscript submission.

A) Statistical test information. Only p-values that are actually shown in the figure panel(s) should (and must) be defined in the legends, all others should be removed from (or added to) the legend. Moreover, we ask for the specification of exact p-values:

- Please note that the exact p values are not provided in the legends of figures 2D, J, M; 5D, 6E, 7E.
- Please indicate the statistical test used for data analysis in the legends of figures 3B, C; 7A.

B) Replicates and error bars:

- Please note that the error bars are not defined in the legend of figure 1A.

C) Data presentation:

- Please note that the scale bar needs to be defined for figures 2G, 6D, G

- Appendix Figure S2A seems to miss the scale bar.

- As a standard procedure, we edit the abstract and title of manuscripts to make them more accessible to our general readership. Please find my suggestions on the abstract below my signature.

- Finally, EMBO Reports papers are accompanied online by

A) a short (1-2 sentences) summary of the findings and their significance,

B) 2-3 bullet points highlighting key results and

C) a schematic summary figure that provides a sketch of the major findings (not a data image).

Please provide the summary figure as a separate file in PNG or JPG format at a size of 550x300-600 pixels (width x height).

Please note that the size is rather small and that text needs to be readable at the final size. Please send us this information along with the revised manuscript.

With kind regards,

=====

Referee #1:

[supports publication in the summary evaluation sheet returned with the report]

Referee #2:

The authors have significantly improved the manuscript by adding valuable data. I consider the revised version suitable for publication.

Referee #3:

The authors have satisfactorily addressed my comments and the manuscript is now suitable for publication in EMBO Reports

=====

Suggested Abstract:

Maternal protein homeostasis and timely degradation of maternal mRNAs are essential for meiotic cell-cycle progression and subsequent embryonic development, but the mechanisms of maternal protein degradation are poorly understood. Here, we show that KLHL8, a substrate adaptor of Cullin-RING E3 ubiquitin ligases, is highly expressed in mouse oocytes and co-localizes with mitochondria. Oocyte-specific deletion of Kihl8 causes oocyte maturation defects and female infertility. ZAR1, an RNA binding protein that is required for mitochondria-associated ribonucleoprotein domain (MARDO) dissolution, is specifically recognized and degraded by KLHL8-mediated ubiquitination. In Kihl8-deficient oocytes, ZAR1 accumulation causes abnormal MARDO and mitochondria clustering, correlating with impaired maternal mRNA decay. Supplementation with exogenous Kihl8 mRNA rescues the degradation of ZAR1 and the dissolution of the MARDO in Kihl8⁰⁰ oocytes. Taken together, our study shows that KLHL8 mediates the ubiquitination and degradation of ZAR1, thus regulating maternal mRNA clearance during oocyte maturation. These findings provide new insights into the roles of the ubiquitin proteasome system during oocyte maturation and establish an interaction network between ubiquitination modification, RNA binding proteins, and maternal mRNA.

The authors have addressed all minor editorial requests.

Zhijia Zhang
Fudan University
China

Dear Dr. Zhang,

I am very pleased to accept your manuscript for publication in the next available issue of EMBO reports. Thank you for your contribution to our journal.

Yours sincerely,
